# The complexity of protein interactions unravelled from structural disorder

**Beatriz Seoane**[1,2,3]*, **Alessandra Carbone**[1]*

**1** Sorbonne Université, CNRS, IBPS, Laboratoire de Biologie Computationnelle et Quantitative - UMR 7238, Paris, France, **2** Sorbonne Université, Institut des Sciences du Calcul et des Données, Paris, France, **3** Departamento de Física Teórica, Universidad Complutense, Madrid, Spain

* beseoane@ucm.es (BS); alessandra.carbone@lip6.fr (AC)

**Data Availability Statement:** The authors confirm that all data underlying the findings are fully available without restriction. All the data is freely available in www.lcqb.upmc.fr/disorder-interfaces/.

## Abstract

The importance of unstructured biology has quickly grown during the last decades accompanying the explosion of the number of experimentally resolved protein structures. The idea that structural disorder might be a novel mechanism of protein interaction is widespread in the literature, although the number of statistically significant structural studies supporting this idea is surprisingly low. At variance with previous works, our conclusions rely exclusively on a large-scale analysis of all the 134337 X-ray crystallographic structures of the Protein Data Bank averaged over clusters of almost identical protein sequences. In this work, we explore the complexity of the organisation of all the interaction interfaces observed when a protein lies in alternative complexes, showing that interfaces progressively add up in a hierarchical way, which is reflected in a logarithmic law for the size of the union of the interface regions on the number of distinct interfaces. We further investigate the connection of this complexity with different measures of structural disorder: the standard missing residues and a new definition, called "soft disorder", that covers all the flexible and structurally amorphous residues of a protein. We show evidences that both the interaction interfaces and the soft disordered regions tend to involve roughly the same amino-acids of the protein, and preliminary results suggesting that soft disorder spots those surface regions where new interfaces are progressively accommodated by complex formation. In fact, our results suggest that structurally disordered regions not only carry crucial information about the location of alternative interfaces within complexes, but also about the order of the assembly. We verify these hypotheses in several examples, such as the DNA binding domains of P53 and P73, the C3 exoenzyme, and two known biological orders of assembly. We finally compare our measures of structural disorder with several disorder bioinformatics predictors, showing that these latter are optimised to predict the residues that are missing in all the alternative structures of a protein and they are not able to catch the progressive evolution of the disordered regions upon complex formation. Yet, the predicted residues, when not missing, tend to be characterised as soft disordered regions.

**Funding:** BS and AC were supported by LabEx CALSIMLAB (public grant ANR-11-LABX-0037-01 constituting a part of the "Investissements d'Avenir" program - reference: ANR-11-IDEX-0004-02); BS was supported by the Comunidad de Madrid and the Complutense University of Madrid (Spain) through the Atracción de Talento program (Ref. 2019-T1/TIC-12776) and partially supported by Ministerio de Economia, Industria y Competitividad (MINECO) (Spain) through Grant PGC2018-094684-B-C21 (also partly funded by the EU through the FEDER program). The funders had no role in study design, data collection and analysis, decision to publish, or preparation of the manuscript.

**Competing interests:** The authors have declared that no competing interests exist.

## Author summary

The Protein Data Bank (PDB) is crowded with proteins that are partially or totally structurally disordered. Nowadays it is widely accepted that Nature uses this disorder to increase a protein's number of possible conformations and interaction interfaces. In this work, we show that the relation between interfaces and structural disorder goes much deeper: the existence of soft structural disorder might be necessary to make an interface. Indeed, interactions with partners take place in the floppy parts of a protein, which means that soft structural disorder might determine the order at which complexes are assembled. Our results are supported by a large-scale analysis of all the crystallographic structures of the PDB and uses no fine-tuning nor learning algorithms.

## Introduction

The paradigm by which the protein function is determined by its three-dimensional structure is one of the basis of Molecular Biology. However, as long as the number of experimental structures increases, it gets clearer that many perfectly functional proteins either lack a well defined structure or they are largely unstructured [1, 2]. These proteins are known as intrinsically disordered proteins or regions, they are fairly abundant among known proteins [3] (and ubiquitous in eukaryotic proteomes [4, 5]) and pathologically present in severe illnesses [6].

The increasing importance of the intrinsically disordered proteins is calling for a reformulation of the structure-function paradigm itself [7, 8], but the biological function of structural disorder is far from being understood. Intrisically disordered proteins are known to enhance the protein's flexibility, and with it, increase its number of possible conformational states. Furthermore, many intrisically disordered proteins and regions possess many disordered interaction motifs [9] that can be used to form alternative complexes and assemblies [10, 11]. All together, structural disorder is regarded as a mechanism to increase the protein promiscuity [12–15] and enrich its functional versatility [16, 17].

This agrees with the fact that disordered regions are often involved in tasks associated to molecular recognition, including the gene regulation, folding assistance or cellular cycle control [2]. Nowadays it is believed that one of the main functions of these disordered regions is precisely to facilitate the binding with other partners (other proteins, DNA, RNA or small molecules), so it is often invoked as a novel mechanism of protein interaction, even though the statistical evidence of this claim is rather scarce [18]. In addition, intrinsically disordered regions often become at least partially structured after binding, undergoing a so-called binding-induced disorder-to-order transition [12], and the adopted folding structures can be different depending on the partner [10, 11].

The vast majority of the large-scale computational studies exploring the biological function of IDRs rely on bioinformatics' disorder predictors based on the amino-acid (AA) sequence to fill the gap between the amount of expected and observed disorder [19, 20]. This fact hampers the identification of universal mechanisms mainly for two reasons. First, because more than 60 different predictors, identifying alternative flavors of disorder (not necessarily mutually compatible), including meta-predictors, have been proposed [21, 22]. And second, because these predictive methods, in their majority, are developed using a reduced set of the total experimental information available about disorder (for instance, the information between the observed and missing regions in the X-ray crystallographic structures of the Protein Data Bank (PDB) [23]). This information often disagrees between the different structures of the same protein sequence (see Ref. [24] and this manuscript), so a question of generality arises. Contrary to all

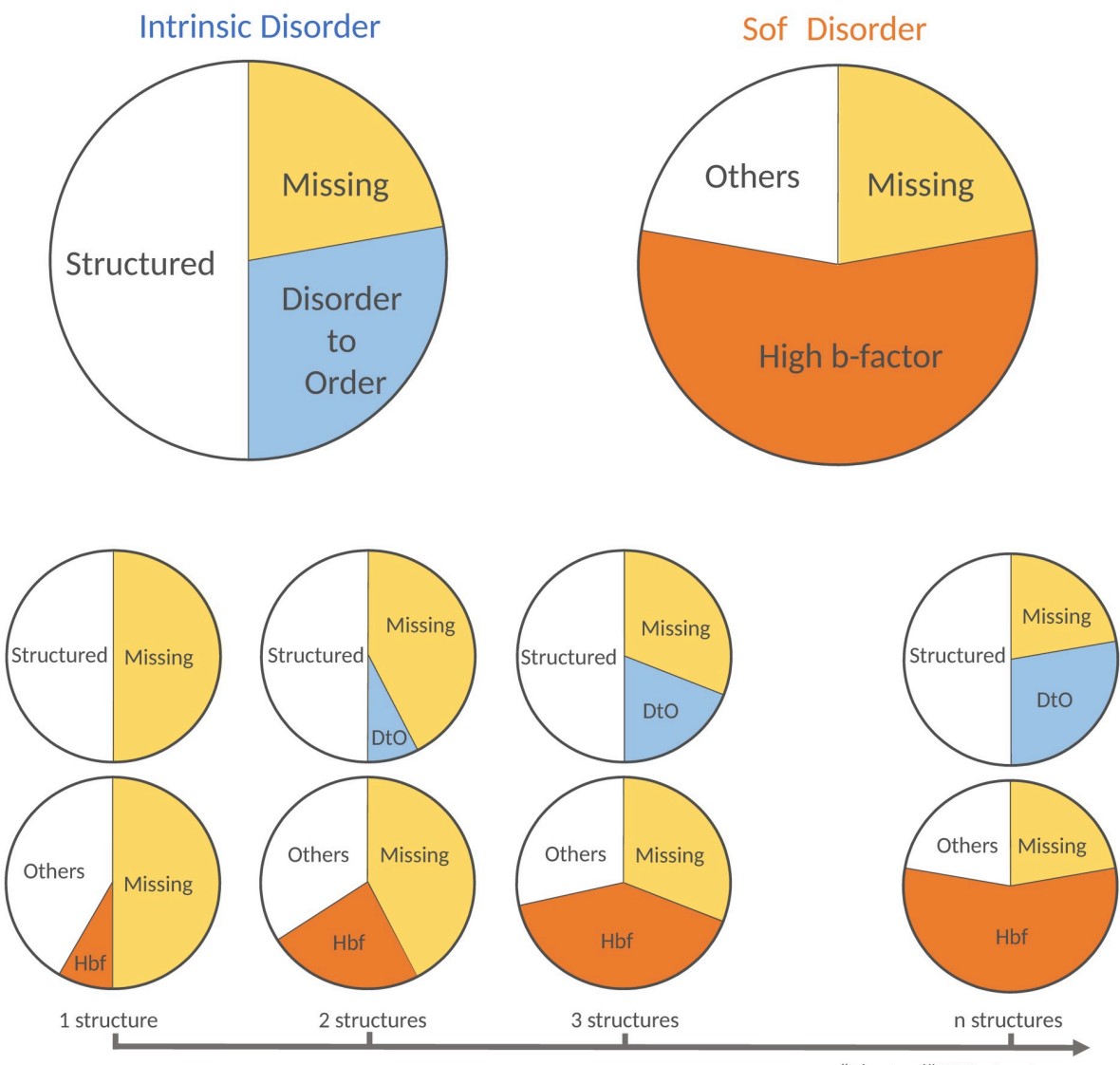

**Fig 1. Intrinsic and soft structural disorder.** We show a sketch of the differences between **A** the traditional intrinsic disorder (defined through the missing residues of the X-crystallographic structure) and **B**, our definition of soft disorder extracted from the high B-factor regions. As more alternative structures of a given protein are included in the analysis, see **C**, the number of always missing residues decreases and a large part of them get eventually structured (the so-called disorder-to-order (DtO) residues). The soft disordered parts grows with the number of structures considered too, see **D**. We observe that the DtO regions are generally characterised also as soft disordered, while missing residues cannot be by definition. The chart sizes are arbitrary, we discuss the relative size and the intersection between both definitions of structural disorder in Fig 6.

those previous approaches, our conclusions rely exclusively on a direct analysis of all the crystallographic structures of the PDB.

In this work, we have analysed all the crystallographic structures available in the PDB, and clustered their chains together when highly similar. Then, we have explored the presence of disorder in these clusters using two alternative definitions: the traditional one, based on the notion of missing residues, and a new definition called "soft disorder" formulated in terms of a high experimental B-factor, see Fig 1. The thermal B-factor quantifies the uncertainty of the atoms positions after the refinement phase of the X-ray experiments. Because of this, a high B-

factor highlights the existence of either *dynamical* disorder (or flexibility) in that part of the protein, or *static* disorder (atoms freeze in positions that are different along the crystal) [25]. Hence, the high B-factor allows us to investigate these two sources of disorder in relatively well structured configurations, as compared to the *hard* disorder, that is, the missing regions where no structure is available. By exploiting the redundancy of the PDB, we looked at the same protein chain in alternative structures of the database and observed residues in the chain that display a relatively high B-factor in at least one of the structures in the PDB. The cumulative high B-factor defines the *soft disorder* for the protein (see Fig 1B and 1D). In parallel, one also observes residues in the chain that are missing in some structures of the PDB and go from disorder-to-order (DtO) in others. The set of these residues together with the missing ones is referred to as *intrinsic disorder* (see Fig 1A and 1C). Fig 1C and 1D illustrate that the definitions of missing, DtO and soft disorder are only meaningful once the information of several structures of the same protein cluster is combined. In fact, we observe, as we will discuss later, that the missing regions that suffer DtO transitions in other structures of the cluster have almost always high B-factor once structured, thus contributing also to the measure of soft disorder. In other words, the DtO regions are independently identified as soft disorder, as we illustrate in Fig 1. Furthermore, we also show that the location of soft disordered regions in a protein is highly correlated with the location of the total interaction interface of the protein with all its partners (either proteins, DNA or RNA). We observe this effect in the DtO regions too, but their relative sizes are much smaller, they tend to cover only a small part of the total interface, even if we observe that they have a similar probability than the soft disordered regions to end up belonging to the interface.

Concerning the interfaces, we observe that the alternative interface regions measured in the different crystals of each cluster progressively add up following a hierarchical distribution of interfaces (see Fig 2) and that the location of soft disorder plays an important role in this organisation. Namely, we observe that, whenever several structures for a given family are available, the union of all the SDRs observed in them predicts relatively well the union of all the IRs, highlighting a direct and simple connection between both mechanisms. Furthermore, we show that the knowledge of the soft disorder of one particular structure gives information about the location of possible alternative interfaces and the order of assembly of complexes. We divide our presentation as follows. We begin with a description of our analysis, followed by the results based on the union of soft disorder. Finally, we discuss the role of the disorder at different stages of complex formation and show several examples where soft disorder predicts the assembly order.

## Analysis procedure

The first step of our analysis is to identify the interface and structural disordered regions in all the X-ray crystallographic structures of the PDB. The interface regions (IRs) are obtained with the union of all the protein-protein/DNA/RNA binding sites. To identify the structurally disordered regions we use two alternative definitions: (i) a classical one, given by the missing regions (MRs) in the PDB structure, and (ii) the *soft* disordered regions (SDR), composed by the residues whose position is poorly resolved in the experiment (and thus having an anomalously high B-factor). A high B-factor is often associated with the concept of flexibility [26], but large thermal motion is not the only possible source for it. In fact, a region having a high B-factor can indicate two distinct situations which are impossible to distinguish from a static picture: the region is (i) floppy or flexible (either it has a mobile, not well defined structure, or it can adopt different alternative conformations [26]), or (ii) its structure is amorphous, meaning by this that, its conformation is essentially rigid (does not change in time), but it is not

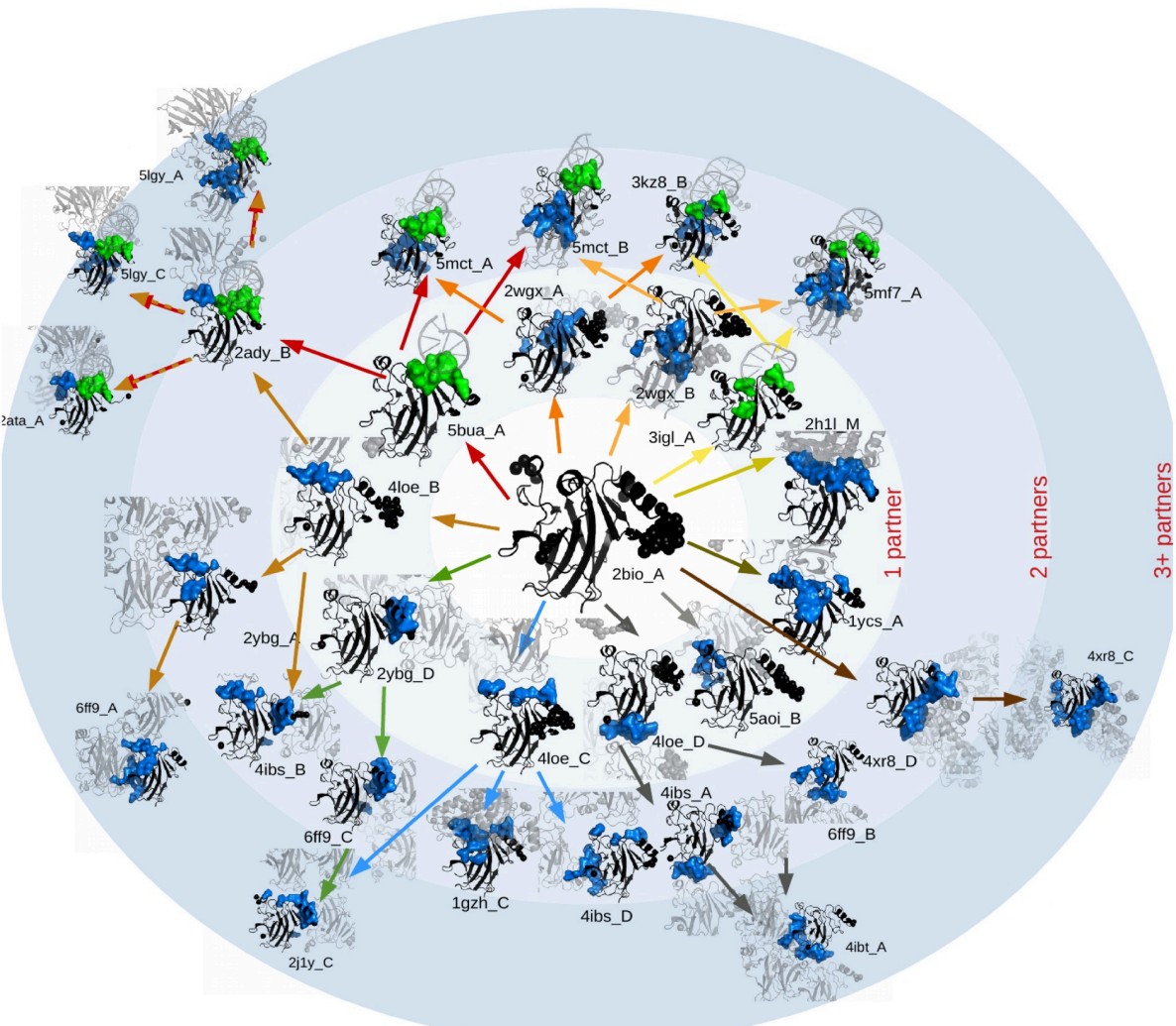

**Fig 2. Hierarchical organisation of interfaces.** We show different structures (PDB ID shown) containing the core domain of the p53 protein (in black) interacting with other partners (in pale grey). Missing residues are modelled as black spheres in the vicinity of their nearest ordered neighbours in the sequence. The interaction interface (on the protein) is displayed in blue if the partner is a protein, and in green, if it is DNA. Structures are ordered laying in concentric circles marking the number of direct partners (within 5Å) of the p53 protein. Cluster labelled as "4ibs_A" contains 196 PDB entries, among which we measured 41 different interfaces (definition later in Materials and Methods —clustering procedure). The graph is generated automatically by comparing the different interfaces. We included here only 32 of them (the PDB ID of each structure is written just next to the structure). For the rest, some were left aside for space reasons (represented here by the radial lines), and the rest because they were qualitatively very similar to the interfaces displayed in the figure. The different interfaces can be placed in a hierarchical graph where each new oriented branch (arrow) represents adding one or more new interfaces to the one shown in the previous structure. Please note that this is just a way of ordering the complexity of the interfaces of the cluster, not a temporal representation of the order of assembly of complexes. A larger version of this figure and additional examples of hierarchies are given in S1 and S2 Figs.

reproducible in the different units that form the crystal (a.k.a. a glass compared with a crystalline solid in Solid State physics) [25]. For this reason, high B-factor has been proposed before as a complementary or alternative measure of protein disorder [27, 28], or as an indirect indicator of its possible presence [20].

The soft definition of disorder aims to soften the classical definition of IDRs, and in particular extend the concept of structural disorder into the "structured" part of the protein. Indeed, residues adjacent to missing regions (in the sequence) tend to have significantly higher B-

factor and we will see later in the analysis that also the MRs that get structured in alternative PDB structures have high B-factor too. Considering that the B-factor varies with the experiment resolution, we always subtract the average of the B-factor, $\langle B \rangle$, and divide it by the standard deviation $\sigma$ both computed using the the entire chain (the B-factor of a residue $i$, $B_i$, is taken as that of its $C_\alpha$ atom). Then, the normalised B-factor (namely b-factor in lower case) of residue $i$ is defined as

$$b_i = \frac{B_i - \langle B \rangle}{\sigma}.\qquad(1)$$

This normalisation allows us to use all the structures in equal foot regardless their resolution or crystal quality. In fact, we observed that limiting the maximum resolution of the structures allowed in the clusters had no systematic effect in the results, see S8 Fig. One could, in principle, use a static threshold in the B-factor for the definition of soft disorder, but this choice hinders the role of the SDRs as we discuss in S3 Fig. If nothing else is mentioned, we will consider that a residue belongs to the SDR if the b-factor is higher than 1, but other thresholds will be discussed.

As a second step, we form clusters of all the PDB structures containing nearly identical protein sequences. Each of these clusters is labelled by the name of its representative chain structure in the form of PDB ID, underscore, chain name; see Materials and methods. We then combine the information of these different experiments by mapping each structure's sequence in the cluster representative's sequence and assigning to each of its AAs the label IR, MR or SDR whenever it was interface, missing or soft disordered at least once in the cluster. We refer to the union of all the IRs, MRs and SDRs of the cluster, as UIRs, UMRs and USDRs, respectively. Furthermore, for the case of the MRs, we distinguish between the residues that are missing in all the structures of the cluster (the intrinsic disordered regions (IDRs)), and the residues that are missing only in a subset of structures (the disorder-to-order regions (DtO)). We illustrate the pipeline of this procedure in Fig 3 and an example of a cluster in mapped in the representative' structure in Fig 4.

In total, we have analysed $N_c$ = 47102 clusters containing 354309 chains extracted from 134337 PDB protein structures. We say that a chain is *bound* if an interface with at least another protein/DNA/RNA chain in the PDB complex is measured, and *unbound* otherwise. 15994 of the total clusters contain one or more unbound chains, but only 5926 of them contain also bound structures. Our clusters are composed of a variable number of chains, having most of them one or less. A more detailed description of the data is given in Section C in the S1 Text and S5 Fig. We found illustrative for our analysis to identify each cluster by its number of different interfaces (NDI). We consider two interfaces as *different* if the number of non mutual AAs in the interfaces is larger than 5% of the sequence length. This choice for the percentage is not critical, qualitatively similar results are obtained with other choices as we discuss in S6 Fig. With this definition, two interfaces that are concentric or slightly displaced, but one significantly larger than the other, are counted as different interfaces. We show in Fig 5A the total number of clusters with a given number of structures (the cluster size) and NDI (in Fig 5B we show the size of the bins used to build that histogram). We will use the same exact binning for the computation of the statistics of clusters of similar size in the rest of the paper. We show a breakdown of the clusters composition (i.e. the containing PDB structures and chains names) and the full list of the clusters in www.lcqb.upmc.fr/disorder-interfaces/.

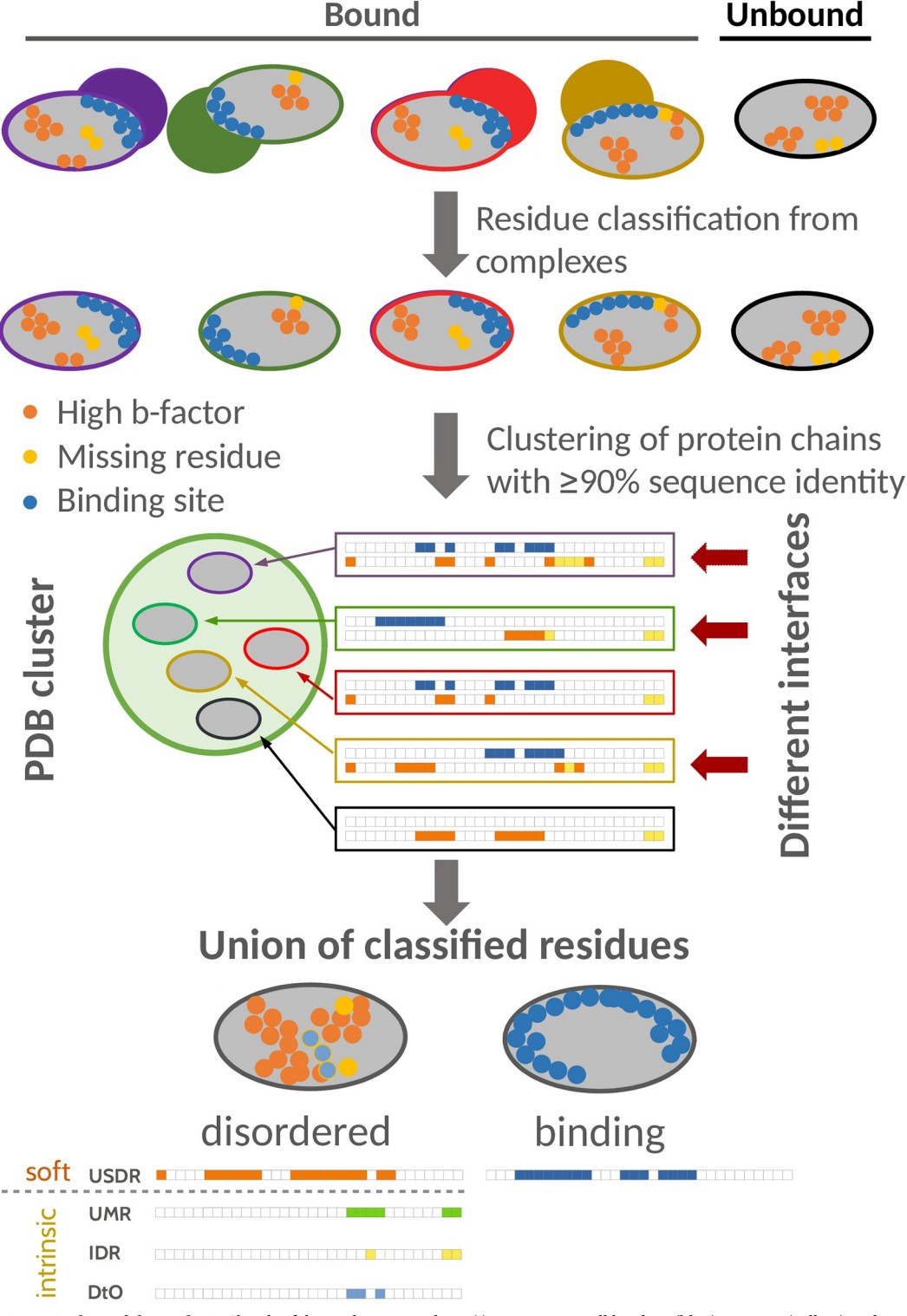

**Fig 3. Pipeline of the analysis.** Sketch of the analysis procedure: (i) We compute all binding (blue), missing (yellow) and high b-factor (orange) residues for all protein chains in the PDB. (ii) We cluster together chains with a common sequence identity ≥90%. Chains without any interface are considered as "unbound" (black structure here), and the rest as "bound". We characterise each cluster by its number of different interfaces (here, the 3 interfaces marked by with an arrow, in a cluster of 5 chain structures). (iii) For each cluster, we compute the UIR as the union of all AAs flagged as binding sites in at least one of the structures of the cluster. For the structural disorder, we compute the USDR as the union of all the AAs having at

least once a high b-factor (in orange), the UMR as the union of all the AAs that were at certain structure missing (green), the IDR as the residues that are missing in all the structures of the cluster (in yellow), and the DtO, as the union of all the residues that were both missing and structured in alternative structures of the cluster (the residues that suffered a disorder-to-order transition, in light blue).

## Results

### Interplay between intrinsic disorder and soft disorder

Once the structural information of all the chains in a cluster is combined, most of the missing residues reported in one or some PDB structures are not missing in the rest. We called the union of these disorder-to-order residues the DtO, and the regions that never get structured, the IDRs. We show in Fig 6A, the median of relative size (with respect to the sequence) among the clusters of a similar number of NDI (that is, the median between the clusters whose size is

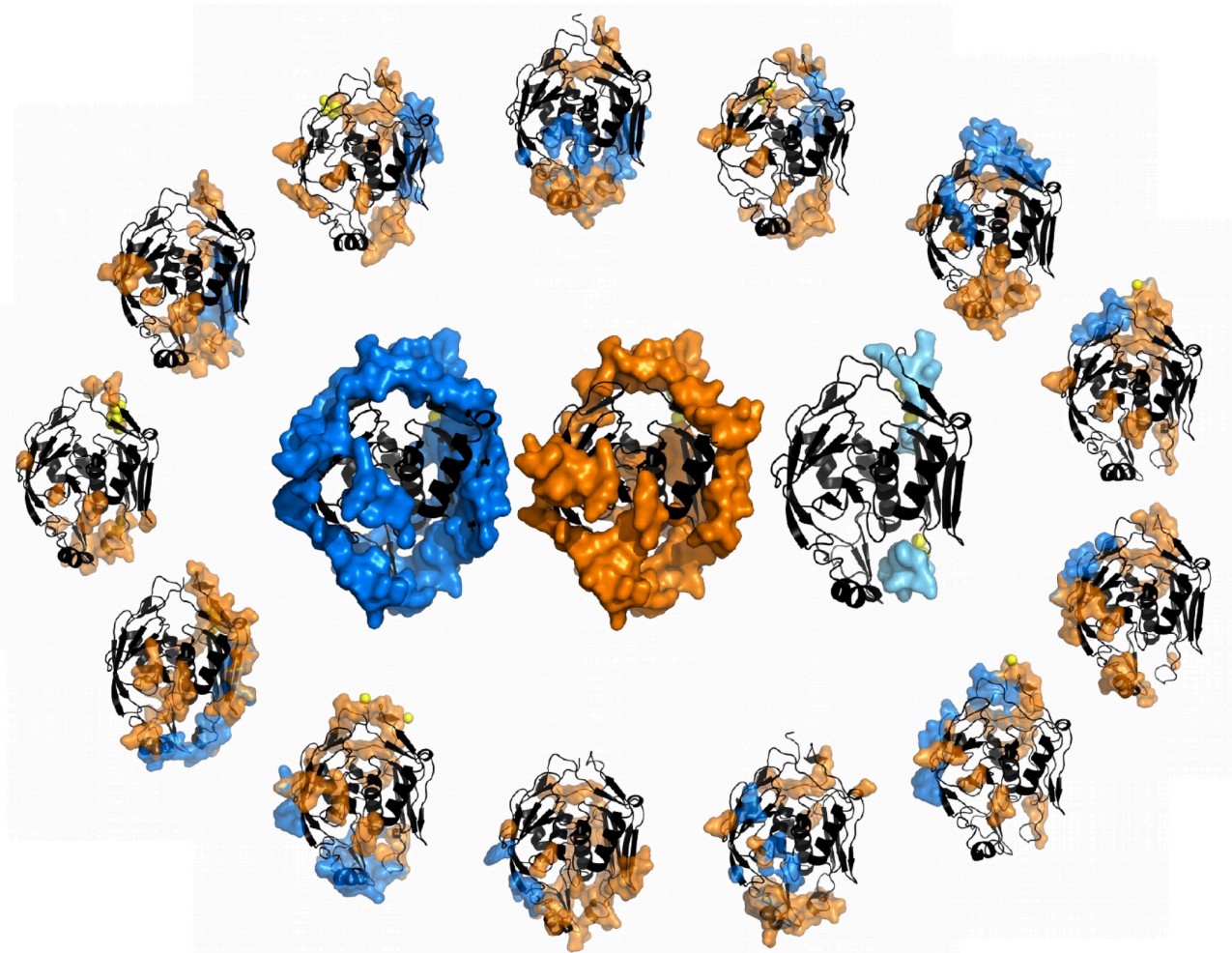

**Fig 4. Example of a cluster of a protein's structures.** Analysis of the cluster 6c9c_A (PDB IDs for the structures are given in S1 Text, Section D). In the external wheel, we show one unbound structure and the chain structures containing the 12 different interfaces of this cluster: SDRs in orange and IRs in blue. Missing residues are modelled as yellow spheres and localised close to those AA in the structure that are direct neighbours in the sequence. In the centre, UIR (blue), USDR (orange) and DtO (lightblue) are mapped on the structure of the cluster's representative. We show an analogous figure including the partners, the b-factor and the binding sites in S4 Fig.

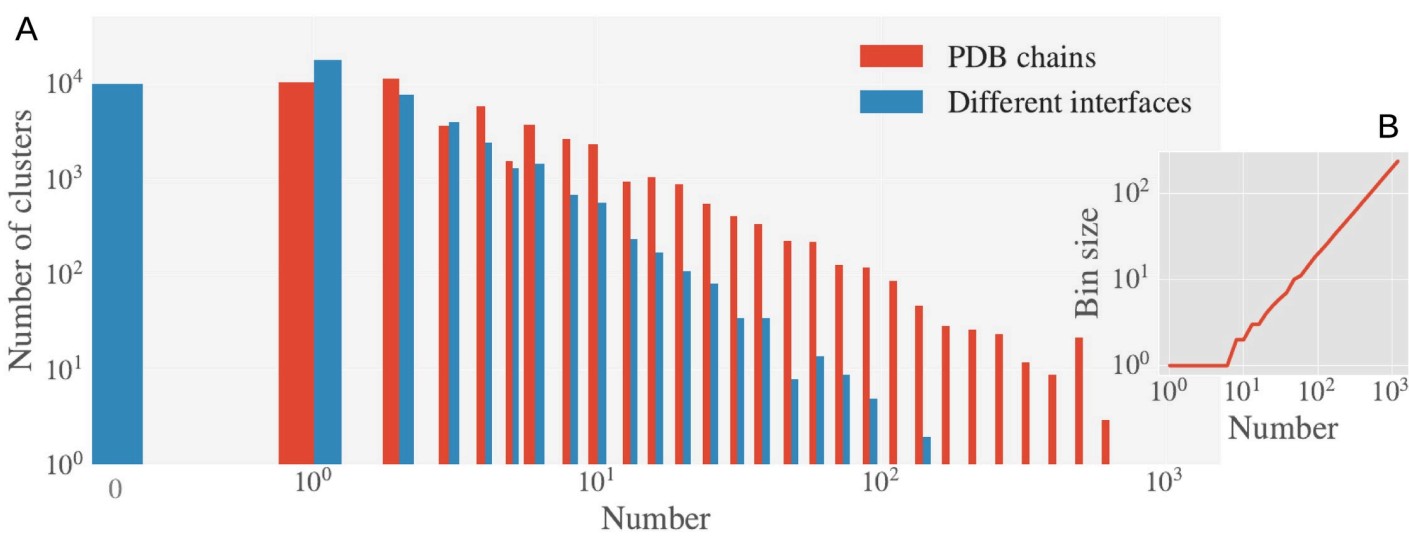

**Fig 5. Histogram of the clusters of structures. A.** Number of clusters with a particular number (number of PDB chain structures, red; and with a given NDI, blue). In **B.** we show the size of the bins in **A**.

included in each bin of histogram Fig 5). In Fig 6B, we show, for each cluster and DtO residue, the fraction of the cluster' structures where the residue is structured (i.e. not missing) that belong to a IR, as function of the cluster size. Despite the current discussion about the role of the DtO creating protein interfaces, statistics tell us that the DtO residues rarely end up belonging to the protein interface when structured. In fact, we observe that the median of the distribution of the frequencies, computed over all structures in a cluster, for DtO residues to belong to the interface is below 5%, in other words, more than a 95% of the times that a missing residue gets structured, it is not located at the interface. Yet, there are large fluctuations and some rare DtOs are almost always at the IRs (about a 14% of the DtO residues are IR in more than half of the structures of a cluster where they are not missing, see light green line). These rare cases have been probably the object study of previous works on disorder-to-order transitions and they might be related to a particular binding mechanism [13, 29]. Also, predicting precisely which missing residues end up in the interface has been the goal of previous studies [4]. Alternatively, we can compare the fraction of DtO residues that belong to the UIR (the union of all the interfaces measured in the cluster) in Fig 6C. In order to see an important fraction of the DtO in the UIR, we need clusters of at least 10 chains. The situation is fairly different when we compare the DtO with the soft disorder, where already 2 structures are enough to see an important overlap (note that having 2 structures is a necessary condition to define a residue as DtO). For this comparison we have considered different definitions of soft disorder, residues with b-factor above 0.5, 1, 2 and 3 (which means that the experimental B-factor is above $\mu + 0.5\sigma$, $\sigma$, $2\sigma$ and $3\sigma$, being $\mu$ and $\sigma$ their mean and standard deviation in the chain, respectively). This figure tells us that almost the 80% of the DtO residues have b-factor higher than 0.5, and almost all of them have b-factor above 1 in at least one of the structures of a cluster of more than 5 structures. This high overlap between being a DtO residue and having high B-factor was already observed in Ref. [28] and was exploited for the definition of the DisEMBL disorder predictor. Furthermore, these results agree with previous studies reporting that the composition of amino acids on residues with high B-factor or in short disordered (missing) regions, appeared to be similar [27]. We can do also a finer analysis within all the cluster structures, and compute the average of the b-factor of the residues that are (or not) tagged as DtO

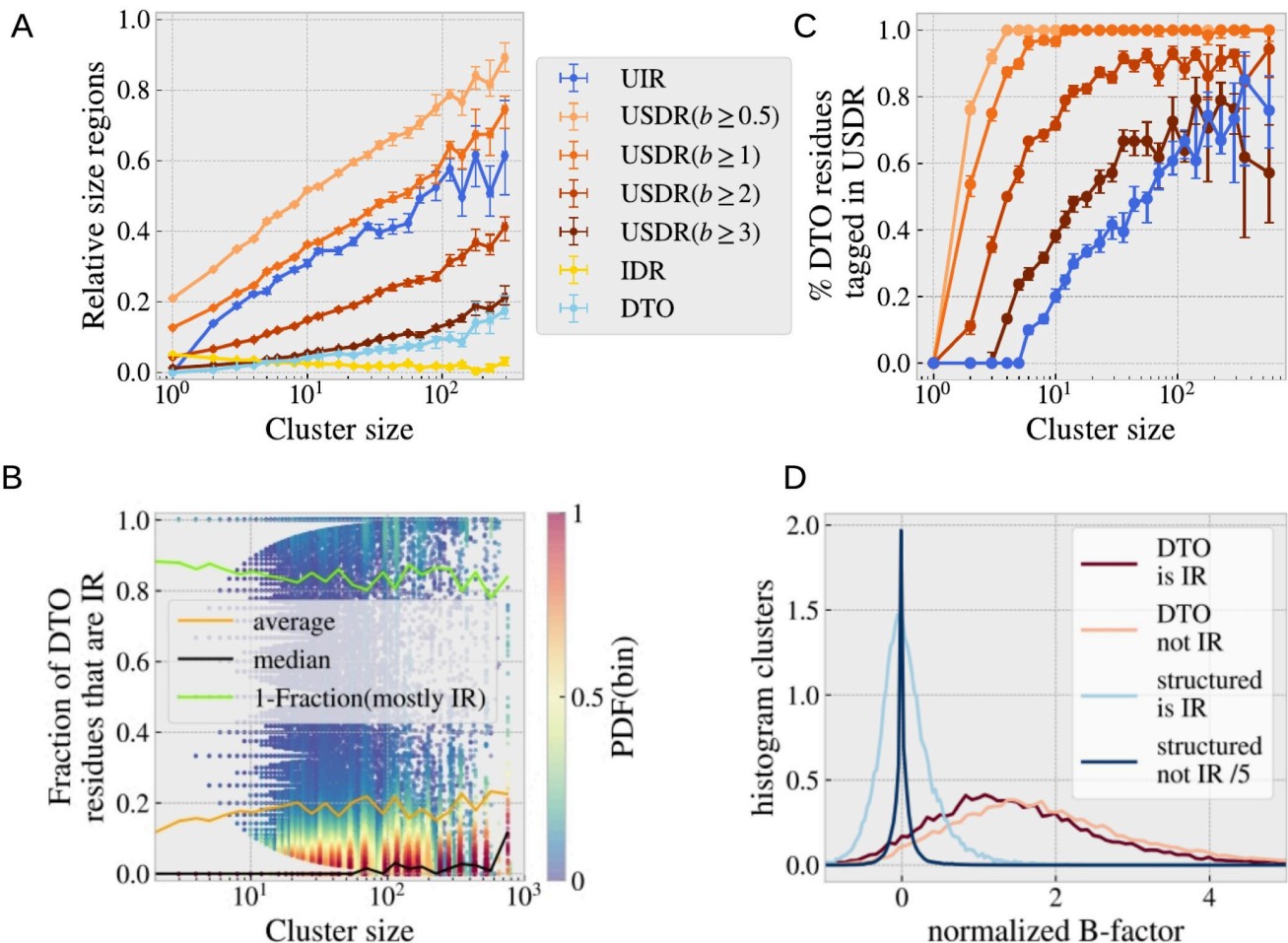

**Fig 6. Statistics of the disorder-to-order regions. A.** We show the relative size (with respect to the sequence length) of the union of the interface regions (UIR, blue), soft disordered regions (USDR, different shades of orange that correspond to different thresholds for the b-factor), the intrinsic disordered regions (IDR, yellow) and the disorder-to-order (DtO, light-blue) regions, as function of the cluster size (averaged using the bins of Fig 5). In **B.**, we show the fraction of the structures in each cluster where the DtO residues belong to the interface, among the total of structures where it is not missing. We show the median and the average of these values using bins of similar cluster size. The large majority of the DtO residues are not part of an interface. We show in light green 1 minus the fraction of these residues that are part of an interface in more than half of the structures of the cluster. In **C.** we show the fraction of the DtO residues that are contained in the rest of the unions of regions in **A**. In **D.** we show a histogram of the mean value of the b-factor in each cluster for structured (never missing) residues and DtO residues, that are (or not) part of the interface region (IR).

in each cluster structure. We also split this data if the residue was (or was not) part of the interface in that particular structure. We show in Fig 6D the distribution of these values among all the clusters in the PDB. The distribution of the b-factor is strongly peaked around zero for the structured regions (a bit below if only IRs are considered), which is nothing but a consequence of our normalisation of the B-factor. However, the DtO residues have clearly much larger values (no matter if they are or not interfaces). These four peaks would be less separated if we had used a static threshold for the B-factor, see S3 Fig).

## Soft disorder and interfaces

The SDRs look qualitatively similar to the IRs in the example of Fig 4, although an AA is not typically both soft disordered and interface at the same structure (binding sites have generally a low B-factor as shown in Fig 6D). This qualitative observation gets more quantitative when

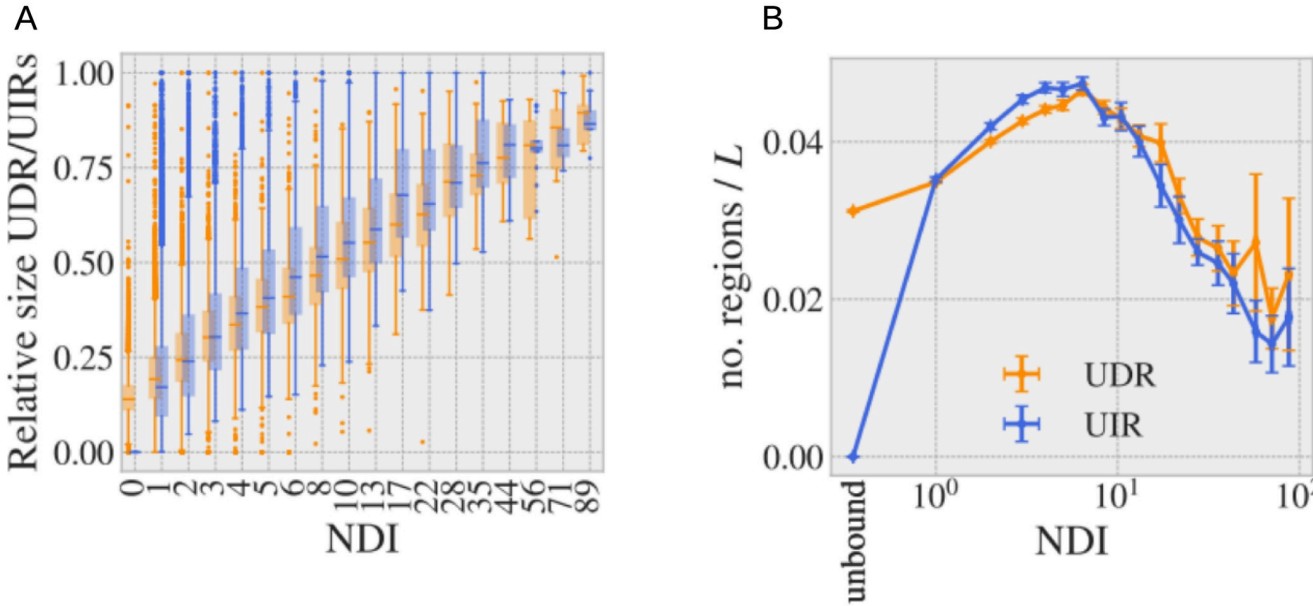

**Fig 7. USDRs vs. UIRs using all the protein chains in the PDB. A** Box-plot of the relative size of the USDR (orange) and UIR (blue) in the sequence, as function of the NDI using the bins of Fig 5 (in the x-label shows the bin's central value). **B** Number of structurally connected regions in the USDR and UIR divided by the sequence length $L$ and averaged over clusters of similar NDI (bins defined as in **A**). The first two points correspond to clusters composed uniquely by unbound structures. The error variables are the standard error of the mean (computed from the bins' cluster-to-cluster fluctuations).

we compare the relative size of the USDRs, $r_D = N_D/L$ with $N_D$ the number of soft disordered residues and $L$ the sequence length, to the relative size of the UIRs, $r_I = N_I/L$ with $N_I$ the number of binding residues, against the NDI (see Fig 7A, in Fig 6D the same data was plotted against the cluster size). The two kinds of regions follow very similar trends as NDI increases, a trend that is destroyed if we randomly displaced the SDRs measured at each experiment, because the USDR quickly covers the entire sequence (see Section F of the S1 Text and S7(A) Fig). This last observation confirms that the positions of the SDRs are not random, but localised in well defined regions, just like the IRs. Furthermore, $r_I$ grows essentially linearly with the logarithm of NDIs, which suggests a hierarchical organisation of the total complexity of the interfaces within the cluster. Meaning that, most of the different interfaces in the cluster enlarge and contain entirely other interfaces in the cluster, so that they can be ordered as following a hierarchical tree organisation as we showed in Fig 2 for the protein P53). Note that, if no such an intricate organisation were present (that is, if the different interfaces were decorrelated one from the other), one would expect a linear growth of $r_I$ with the NDI. The same exact behaviour is observed for the SDRs, which anticipates some connection of the soft disorder with the interface hierarchy.

We can also compare the average number of distinct structurally connected regions in the USDRs and UIRs. To do so, we recursively group together all the AAs assigned as SDR/IR located within a distance (in the three-dimensional structure) of $\leq 6\text{Å}$ to at least one of the other members of the group. As we show in Fig 7B (and explain in Section F of the S1 Text), the number of SDRs and IRs (normalised by $L$) follows, again, a surprisingly similar behaviour once averaged over all the clusters with a similar NDI: they both grow with the NDI up to approximately 6, moment at which regions start to superpose thus decreasing the number of regions. Quite interestingly, a random displacement of the groups of SDRs in the USDR or a

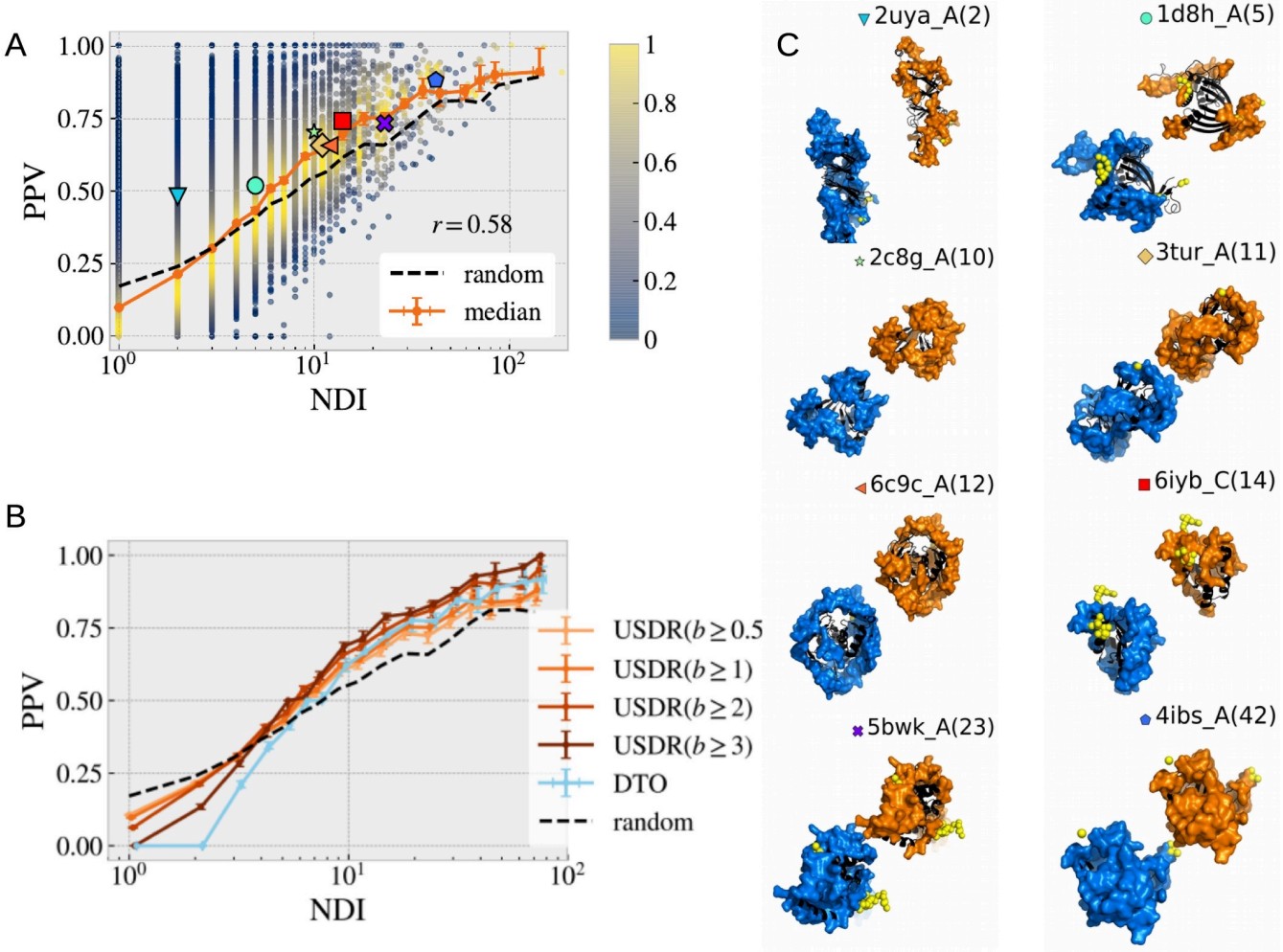

**Fig 8. Correspondence between the location of USDRs and UIRs.** We compare the sequence location of the USDRs with the location of the UIRs. In **A.**, we show the probability that a USDR (defined as $b \geq 1$) residue is also included in the UIR for all the clusters in our database (dots). The dot colour marks the density of points in the PPV region in eac h bin. In orange error-bars we show the median among clusters obtained in each bin. The dashed black lines shows the median among the same clusters of the 'no correspondence' expectation. In **B.**, we show the PPV medians by bin obtained using different definitions for the USDR (in different shades of orange) and using the DtO residues (in light blue). In **C.** we show several snapshots of the protein structure of 8 clusters where the UIR and USDR in the cluster is shown as blue and orange surfaces, respectively. The PDB ID of the cluster representative and the cluster's NDI are shown on the top, together with a big-colour point that allows us locate the PPV metrics of **A**.

total randomisation of the disordered sites of the USDR along the sequence, lead to either lower or totally different curves, respectively, as shown in S7(B) Fig.

In average, the UIRs and USDRs display a remarkably similar statistics as the clusters' NDI grows, both in terms of relative sizes and number of structurally connected regions. It is then natural to wonder to which extent both regions overlap. We compute the probability that a residue in the USDR is part of the UIR (the so-called positive predictive value, PPV, defined in Materials and methods). In Fig 8A, we show the value obtained for each of our clusters as function of the cluster's NDI. We show in orange the median of the clusters of similar NDI. If we compare this line, with the expected value if both regions were totally unrelated (black line), one concludes that observing a SDR at a certain structure of the cluster, increases the probability of measuring also an interface in a structure at the same exact site. The same qualitative behaviour is observed in all the metrics tested, see in the Materials and methods and S9 Fig

The USDR shown in Fig 8A was obtained using the $b \geq 1$ threshold, but other values or even the DtO regions could be used for the analysis, as we show in Fig 8B. Again, the same gain over a random guess is observed in clusters with more than 3 different interfaces, being quite remarkable that the higher the threshold, the higher the PPV is. An important presence of DtO precisely in the binding sites was already shown through a large scale study of a similar design in Ref. [30], but soft disorder extends this idea to much larger regions.

It is commonly mentioned in the Literature that missing residues get structured (or undergo a disorder-to-order transition) to create interfaces. Yet, we can quickly see that the correspondence between the soft disorder and the interface sites is more general than just the DtO residues forming interfaces. In fact, if we remove completely from the USDR the regions the amino-acids that were reported missing at least once in the cluster, we observe essentially the same results, as we show in S10 Fig. In this sense, soft disorder alone is as correlated with new interfaces as the DtO regions (recall the light blue DtO PPV curve in Fig 8B), showing that they both play a very similar role forming protein interfaces. Yet, let us stress that the information about both kinds of disorder must be combined if we want to cover all the possible interfaces: DtO regions are smaller but yet they tend to highlight regions where interfaces are later found.

Furthermore, while the USDR ($b \geq 1$) have a similar PPV than the DtO, the soft disorder covers much larger regions of the protein, reaching sizes comparable to that of the UIR. We show in Fig 8C, several examples of the USDR and the UIR mapped in the cluster representative's structure for clusters with different NDI. These examples strongly suggest that USDRs and UIRs cover essentially the same AAs in the protein. We can test this correspondence through a ROC curve, that is, comparing for each of the clusters, the portion of the UIR covered by the USDR (the so-called Sensitivity, Sen), versus the fraction of USDR residues that are not UIR with respect to the number of sites that do not belong to the UIR (which is 1 minus the Specificity, Spe). See in the Materials and methods for more details. We show in Fig 9A the values obtained for each cluster containing more than 8 different interfaces (the dot colour marks its NDI), together with the median between clusters using 20 bins equally spaced in the $x$-axis, showing a clear better performance than a random guess. In Fig 9B we show the medians obtained considering different thresholds of NDI. For $NDI \geq 1$, the match between USDR and UIR is worse than random, which agrees with the observation that interfaces have rarely a high b-factor (note that, as shown in Fig 5A, the large majority of the clusters have just one NDI). However, the higher this threshold is, the better the correspondence between both regions is. Qualitatively similar results are obtained using other definitions for NDI, as we show in S6 Fig. One cannot forget that the PDB is very incomplete, which means that a higher NDI means also a better knowledge of the protein interaction network. This fact strongly suggests that soft disorder can be used as a measure of protein interface propensity. We can repeat the same analysis using other definitions of structural disorder (see Fig 9C for all clusters with interfaces and Fig 9D, for only clusters with NDI $\geq$ 8). Despite having a much higher PPV, the USDR obtained with b-factors above 2 or 3 are not good to describe the totality of the interface regions. The same happens with the DtO, probably suggesting that these very disordered regions are used to form a very particular type of interface. On the other hand, a lower threshold for USDR ($b \geq 0.5$) increases notably the sensibility without sacrificing equally the specificity, despite having a lower PPV, which makes this observable a good candidate for determining the interface propensity. Yet, we find this measure hard to use for interpretation because the different structural regions quickly superpose with the NDI, thus making difficult the visual identification of the different regions of interest. For this reason, we decided to keep the $b \geq 1$ case as the predominant example for the paper.

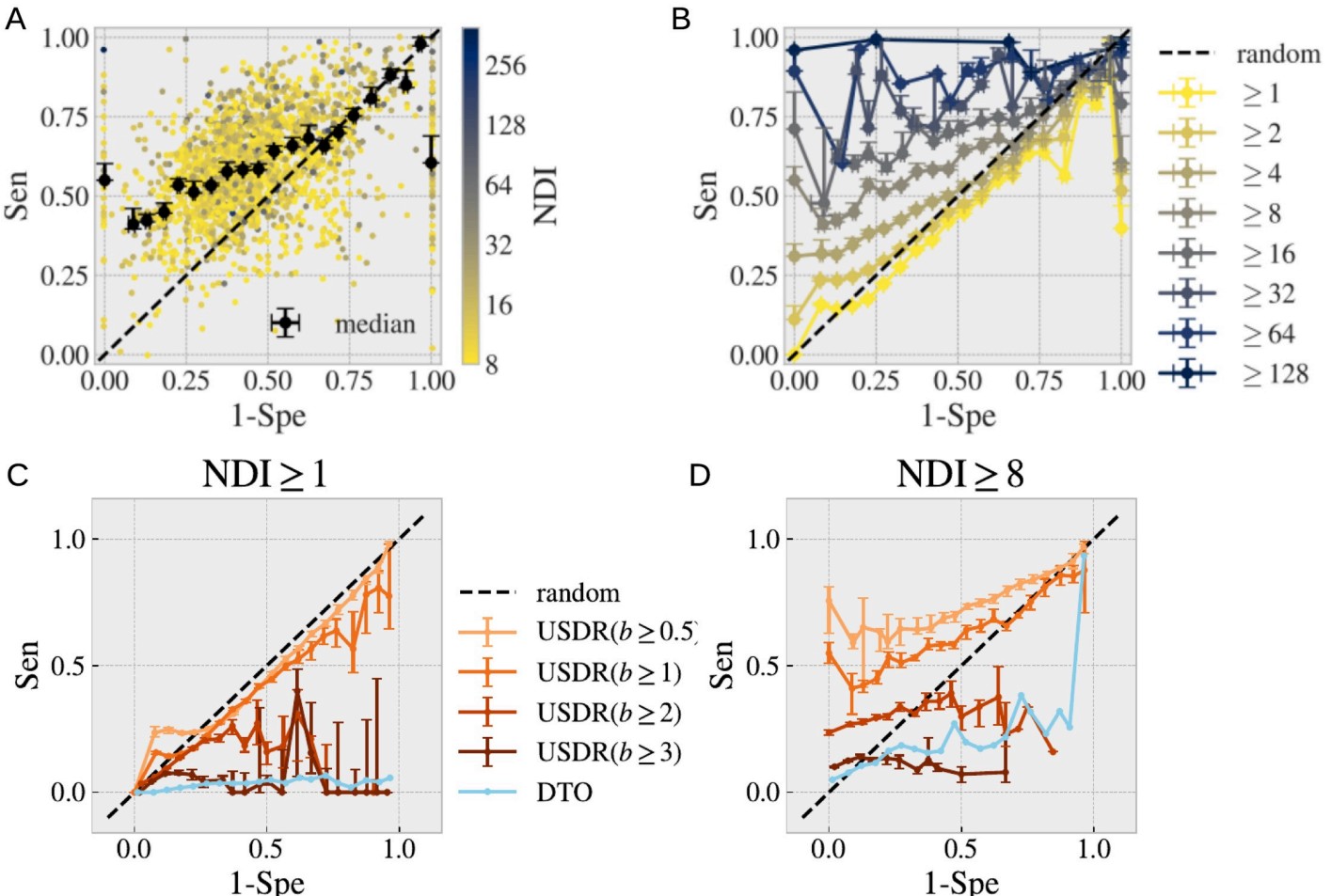

**Fig 9. Interface propensity of the union of soft disorder.** We test how well the USDR (for $b \geq 1$) describes the UIR. For this, we compute the ROC curve (Sensitivity against 1-Specificity) for all our set of clusters. In **A**, we show all the clusters (dots) having 8 or more NDI (the dot colour marks the cluster's NDI). In black, we show the median of the Sensitivity obtained using 20 evenly spaced bins in the Specificity. In **B**, we show this median for other choices of the threshold in the cluters' NDI. We show the medians obtained with different definitions of structural disorder (different thresholds of b-factor to define the USDRs and the DtO) for NDI $\geq 1$ (in **C**) and for NDI $\geq 8$ (in **D**).

The observed correlation between high b-factor residues and binding sites, is in the line of what observed before in an indirect way, that b-factors are useful to predict interfaces with other partners (see, for instance, Refs. [31, 32]), or even to discriminate between protein binding interfaces and crystal packing contacts.

## Disorder and protein assembly

We showed in Fig 9B that several different interfaces were needed to observe a good match between the USDR and the UIR. However this is only true if our clusters do not contain unbound structures (as it is the most common case). In Fig 10A, we show the same ROC curves, but this time using only the clusters that contain one or more unbound structures. In this case, the match between both kinds of regions is better than random even for 1 NDI. This fact suggests that soft disorder might have a role in allowing the next step of complex assembly. If it were the case, an USDR computed only using unbound structures, should still carry information about the location of the interfaces observed in the rest of the cluster.

                    

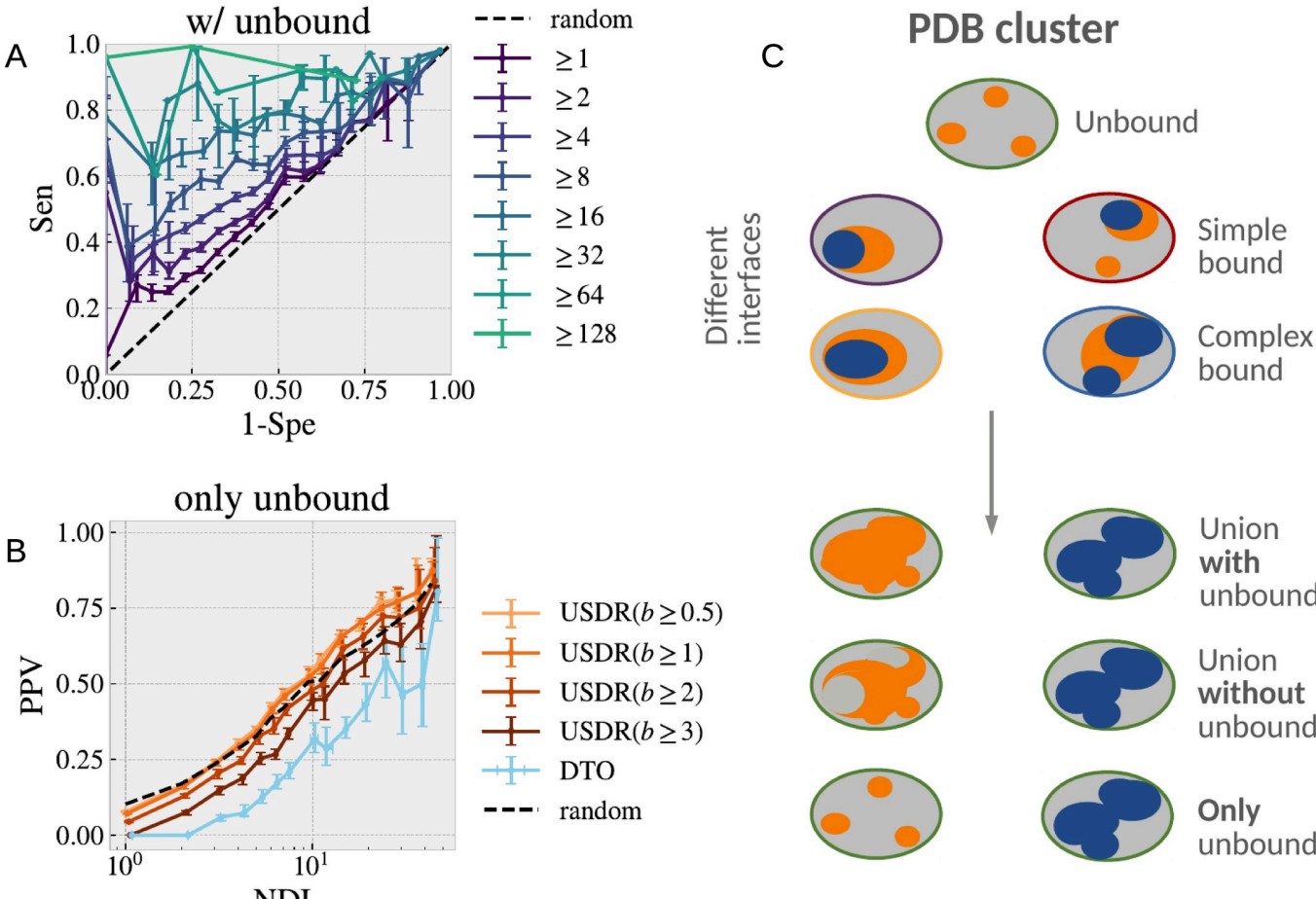

**Fig 10. Disorder leads the assembly path. A** We show a ROC curve analogous to that of Fig 9B, but this time computed only using the clusters containing unbound forms. With this change, we start to see a non-trivial correspondence between the USDR and the UIR already from clusters with 1 interface. In **C**, we show the percentage of the USDR residues (for different definitions of soft disorder) in the unbound forms that end up part of the interface in the bound forms of the cluster, as function of the NDI. We compare these results with the PPV expected for a random guess. **C** Sketch of our hypothesis of the role of SDRs at the different steps of assembly in a cluster of 4 NDIs. Unbound structures present several disconnected SDRs (orange) that mark the location of the possible IRs (blue) accessible from that structure (we call them simple bounds). In the same way, the SDRs observed in bound structures, predict the possible IRs of that complex with new partners (what we call complex bound). Below, we show a sketch of the union of SDRs in clusters with, and without unbound structures, and using only the information from unbound structures, compared with the union of all the IRs in the cluster.

In Fig 10B we show the PPV of the union of the different definitions of disorder computed only using unbound structures and compared to the UIR of the entire cluster. For the sake of comparison, we include also the expected PPV if there were no connection at all. Again, the USDRs obtained for $b \geq 0.5$ and 1 in the unbound have a tendency to end up at the interface.

These two observations suggest that the SDRs in the unbound forms of a protein carry information about the interface sites of this protein with other partners, but also that the SDRs of the bound forms must have a similar function (the PPV of the total cluster USDR is higher than that of the USDR of the unbound). This makes us hypothesise that the USDRs in the unbound forms tells us about the propensity to form simple interfaces (interactions between the unbound protein and a limited number of partners), and that the USDR measured in the bound forms, would highlight the regions at which new interfaces will be placed when the complex interacts with new partners. We illustrate this idea in the sketch of Fig 10C.

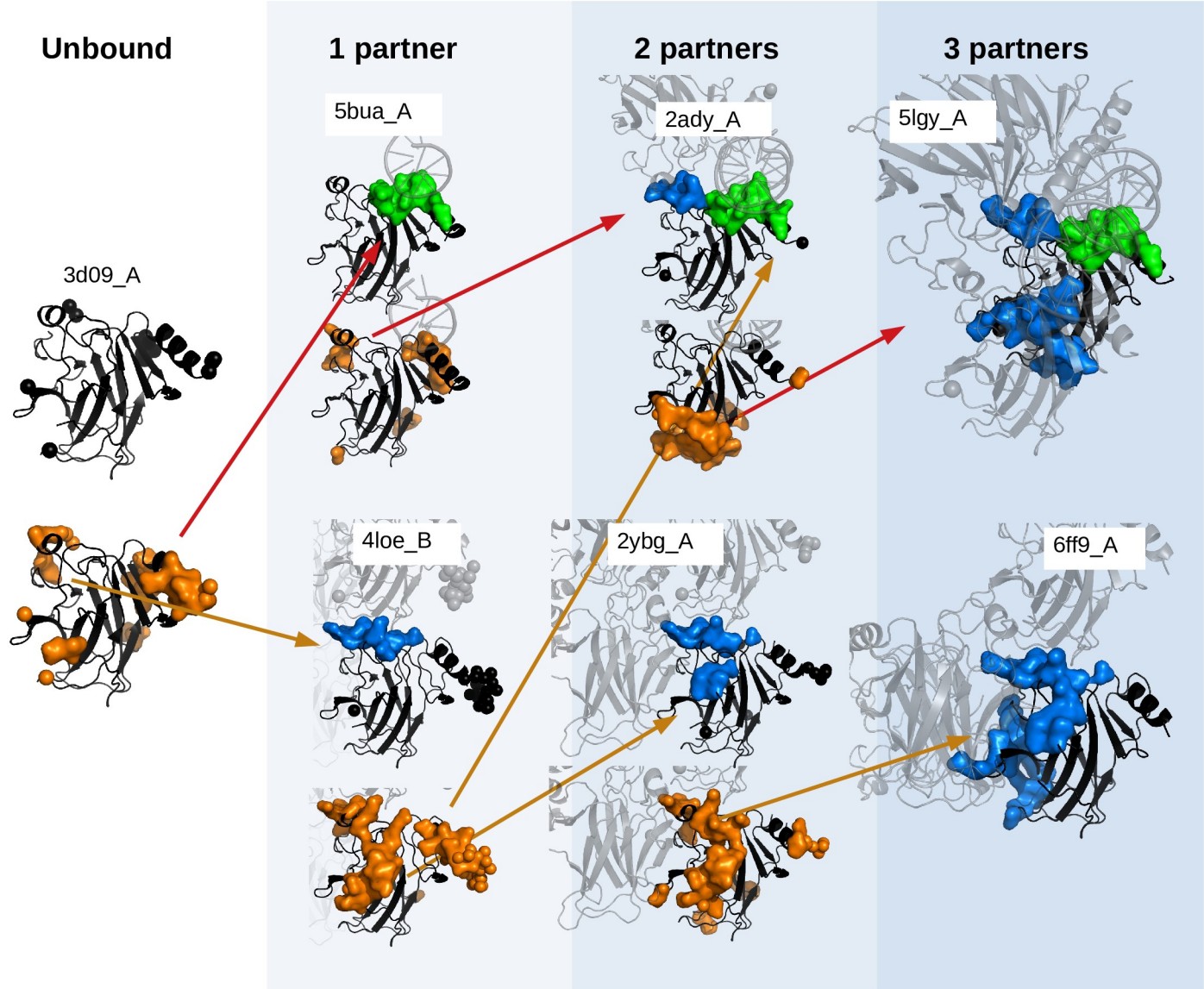

**Fig 11. Progressive change of the disorder.** In **A**, we show the SDRs (orange) of all the unbound structures of the cluster 4ibs_A (the core domain of protein p53) at the sides of the figure, together with the union of all of them in the large structure in the centre. In addition, we include several structures (in black) containing 10 different interfaces with a single direct interaction partner shown in Fig 2 (partners' structure in pale grey) in the external wheel (interfaces with proteins in blue and with DNA in green), showing that interfaces occur in regions that were characterised as disordered in the unbound structures. In **B**, we show the progressive change of the SDRs upon binding along the red and orange branches of the tree in Fig 2. For instance, the SDR of one of the unbound structures in the PDB allows us to predict the interfaces of this protein with other partners (DNA in PDB ID 5bua_A and another p53 protein in PDB ID 4loe_B). Again, the SDRs measured at these complexes show the location of the new interfaces with a second partner (PDB IDs. 2ady_A and 2ybg_A). This situation is also observed when a third partner is considered (PDB IDs. 5lgy_A and 6ff9_A). The background colours refer to the different concentric circles in Fig 2.

Fig 10 proves that this hypothesis is valid at the first step of complex assembly (from unbound to bound forms), but checking it statistically at higher steps of protein assembly would require, not only families of structures of similar proteins, but also clusters of similar complexes' structures, which gets much harder at a technical level and will be tackled in future works. Yet, we test this idea in some particular examples of our dataset. For example, we show in Fig 11 some of the structures of the cluster 2c8g_A whose USDR and UIR were shown in Fig 8C. In the centre of Fig 11A, we show the union of the SDRs observed in the two distinct

unbound structures of the cluster (2c8b_X and 1uzi_A), and the different simple bound structures observed in the cluster (just one partner, either one protein or a compact complex). We can see that the USDR of the unbound reflects to a good extent the location of the new IRs. However, see Fig 11B, the SDRs observed in the bound structures change drastically their the location. In fact, they appear at the same regions where we observe interfaces of the same complex with other new partners. Other examples of progressive change of SDRs are included in S11 Fig.

The change of the SDRs at different steps of complex formation highlights some kind of allosteric phenomena, by which the interactions with partners in a particular region, changes the physical structure at a distant part of the protein. Also, this mechanism seems to play a role in the order at which complexes are assembled. A similar mechanism of step-wise targeting and sequential assembly via a binding chain reaction was previously hypothesised in Ref. [33], but it is the first time, to our knowledge, that it is (partially) validated statistically. Furthermore, we can test the specific idea of soft disorder leading the complex assembly, by looking for biologically verified orders of assembly of complexes where we could find all the intermediate structures in the PDB. Considering that the disorder depends on the crystallised structure of the complex, it is important to check that the complete (and not a partial, which is the common case) structure of the complex is available. We have manually identified only two different processes where this was possible. The first, taken from Ref. [34], concerns the verified hierarchical assembly $A > AA' > AA' A'' A''' > AA' A'' A''' EE'$, with $A$ and $E$ two different protein chains. We show in Fig 12A the measured SDRs and IRs in structures at the 4 different steps of the assembly, showing that IRs appear roughly in the same areas where SDRs were observed in structures of the previous step. In Fig 12B, we show the second example discussed in Ref. [35]. According to Ref. [35], homodimers combine preferentially in two forming a structure with D2 symmetry. We show the SDRs in one of the dimers discussed in that paper and compare with the IRs measured in two complexes with that geometry, showing a surprisingly good agreement. This last example is illustrative, because it was pointed out in previous large-scale studies that structural disorder is particularly abundant in symmetric homodimers [30], and the purpose of that disorder could be precisely to assembly correctly these D2 symmetry complexes.

## Discussion

We have shown evidences that the presence of soft disorder increases the propensity of a protein to form interfaces with other partners in a particular region, and this seems to be valid for all kinds of interfaces. This observation explains why the b-factor seems to be useful for protein interface predictors [31, 32, 36]. We also observe that intrinsic disorder has a tendency to end up in interfaces when suffering disorder-to-order transitions, as reported in previous works [30], but its presence in the interface is much less important than that of the soft disorder. Furthermore, we have observed that DtO residues tend to be classified as soft disorder (in agreement with the previous observation that short disordered and high b-factor regions tend to have the same AA distribution [27]), which means that the connection between short disorder and binding regions [30] is contained in our definition of soft disorder. In fact, we find no particular improvement on the estimation of the interface propensity when we combine intrinsic and soft disorder.

Even if the missing residues are not particularly useful to predict interfaces, disorder predictors could be if they manage to predict correctly the DtO (because predictors give access to sequences not resolved experimentally). With this purpose, we have considered three different disorder predictors: IUPred2A [37, 38], SPOT-Disorder2 [39] and DISOPRED3 [40] and

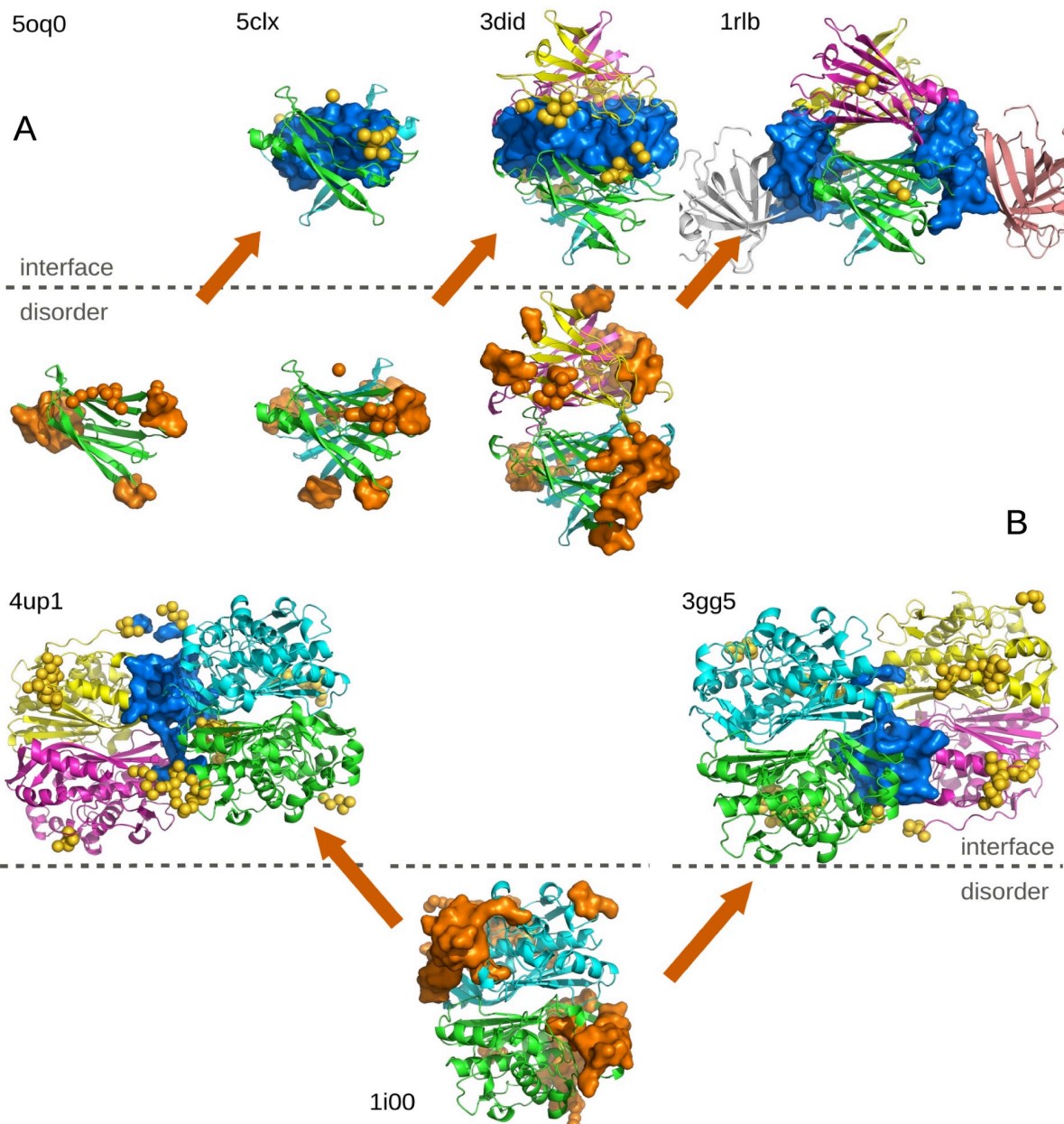

**Fig 12. The role of disorder in the biological order of assembly.** Evolution of the SDRs (orange) and IRs (blue) at different steps of two verified assembly processes. **A** The assembly of the retinol binding protein complexed with transthyretin (1rlb). **B** The assembly of the two identical dimers of the human thymidylate synthase (1i00). PDB IDs are reported for each complex.

compared their predictions with our interface, soft and intrinsic disorder measures. The first drawback we encounter is that, since the AA sequence of all the protein structures of each cluster is essentially identical, and all these three predictors are sequence based, the predictions are totally blind to the progressive changes of disorder upon hierarchical binding. This also means that we only need to compute the prediction of each of these 3 predictors for the representative sequence of each cluster. Then we can compare the predictions with our analysis. In Fig 13A and 13B, for each predictor, we show the PPV and the Sensitivity of these predictions

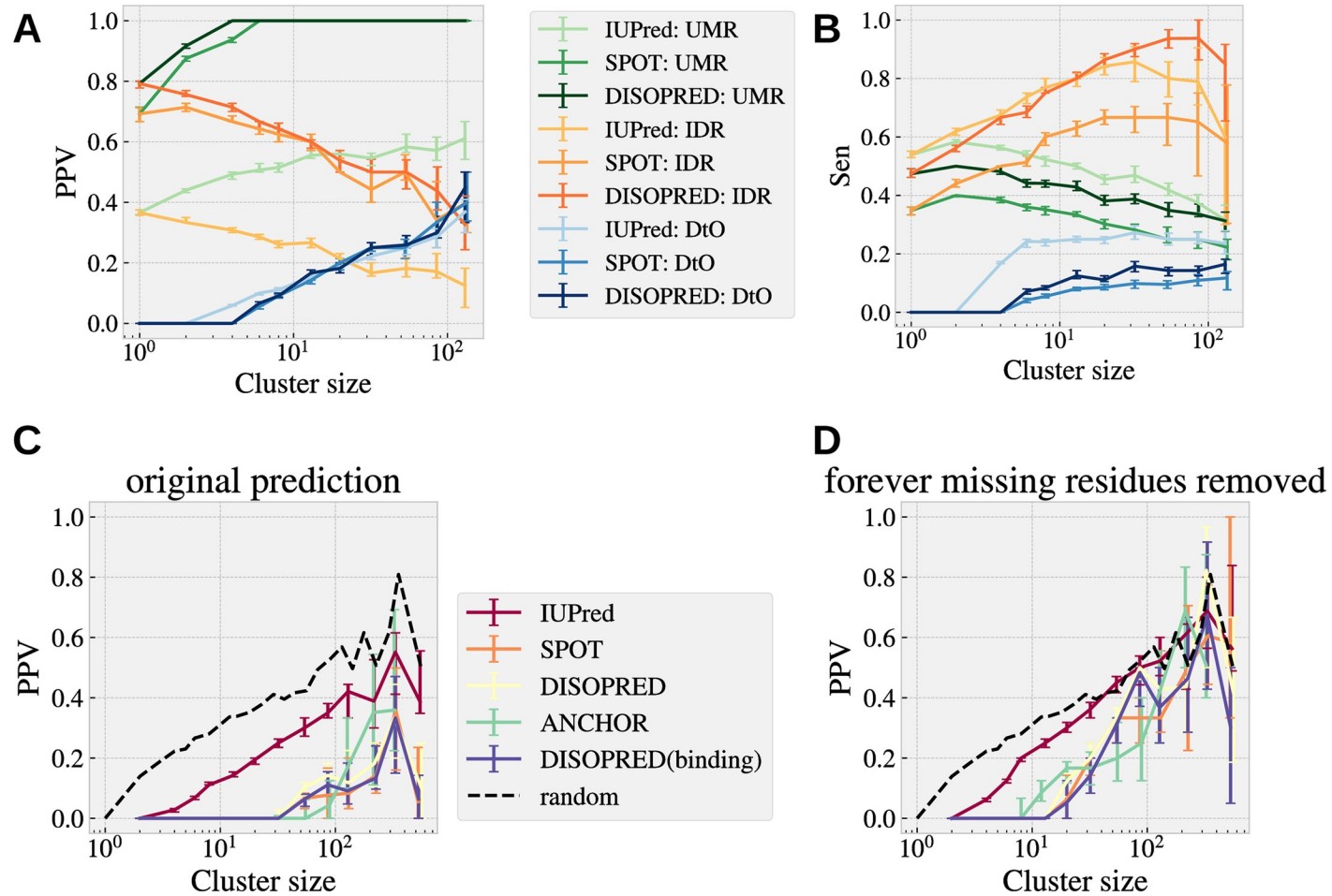

**Fig 13. Disorder predictors between IDR and DtO.** For each cluster, we systematically compare the disorder predictions of 3 intrinsic disorder predictors IUPred2A [37, 38], SPOT-Disorder2 [39] and DISOPRED3 [40] with respect to the union of all the missing residues (UMR), the forever missing residues (IDR) and the union of all the disorder-to-order residues (DtO). We show the cluster's median of the PPV as function the NDI in **A**, and of the Sensitivity in **B**. In **C** show the probability that a predicted missing residue is part of the UIR in the cluster, and in **D**, the same probability once all the IDPs have been removed from the total sequence length. For these last two figures, we have introduced predictions of missing residues with protein binding annotations, in particular, we consider ANCHOR [37, 38] and DISOPRED3 [40].

compared with (i) the union of all missing residues (UMR) observed in the cluster, (ii) the forever missing regions (what we called the IDRs) and (iii) the DtO regions as function of the NDI. Data shows that all 3 predictors are mostly optimised to predict the IDPs, and they only predict a relatively small fraction of all the DtO residues. This fact anticipates that one can hardly use these predictors to determine the interface propensity, but yet they are useful to estimate the total lack of interfaces in the region. Still, the residues wrongly predicted (those that were actually structured in our dataset), are in their large majority part of the USDR, as we show in S12 Fig. One can see that in fact, predicted disorder residues tend to have large b-factor as shown in S3(C) Fig. Once confirming that being characterised as soft disordered is correlated with finding an interface in that site, it is normal to wonder if it happens the same with the predicted disordered residues. To check this, we compute the overlap between these predictions and the UIR. We show in Fig 13C the fraction of predicted disordered residues that end up in the UIR (for clusters with at least one residue predicted disordered) as function of the NDI. In this case, we also include predictions for disorder-to-order interfaces obtained

with ANCHOR [37, 38] and DISOPRED3 [40]. As expected, in all the cases, the PPV is worse than the expected value for completely decorrelated observables (notably even for predictors of disordered regions with annotated protein-binding activity). This fact highlights something trivial, that interfaces do not occur systematically on forever missing regions. Yet, even if these regions (our IDRs) are removed from the total sequence (see Fig 13D), we continue observing a negative correlation between both measures or at maximum, we obtain values compatible with no correlation at all between both observables. Note that this observation does not tell that predictors of binding affinity in disorder regions are not accurate, only that they are not good to predict all kinds of interfaces. In summary, these results tell us that predictors for soft disorder (or high B-factor, as for instance, Refs. [27, 41, 42]) might be much more useful to estimate the interface propensity than intrinsic disorder predictors, which are not adapted for this task. Yet, the could be combined with the former to improve the prediction of predict where interfaces cannot be located.

We end up stressing that we have combined all the crystallographic structures available in the PDB for essentially the same protein, to conclude that interfaces occur preferentially in regions that are characterised as disordered in alternative crystals. Our analysis does not include any kind of fine-tuning of parameters nor predictions from learning algorithms, which both attaches a great confidence to the generality of the results and leaves great room for improvement. Despite the fact that the database is very incomplete, and thus most of the soft disordered regions should seem random in our analysis, we still observe a significant increase of the interface propensity associated to the soft disorder. Yet, we want to stress that, interactions with small molecules have been completely ignored in our analysis (it will be subject of future studies), and these interfaces could probably be an important part of the false positives reported. Indeed, we expect the binding with small molecules to occur precisely in soft disordered regions, because they need to adjust easily. Furthermore, small molecules' binding sites are normally located on the border of protein-protein interfaces, just where we find the soft disordered regions.

We also observe that the different interfaces measured in the big clusters follow a hierarchical organisation and that structurally disordered regions are related to it. Indeed, we show that the location of the structurally disordered regions changes upon progressive binding probably controlling the order of assembly of complexes, as it was theorised in previous works [33]. Thus, our work provides a transverse knowledge to sophisticated models of protein-protein interactions considering, at the two extremes, proteins in dynamic equilibrium that find partners by selection of "minor" species and models that consider partners found among "major" species and involving conformational changes destined to stabilise the complex. Finally, intrinsically disordered proteins are known to fold differently upon binding to different targets [10, 11] and our current analysis is completely blind to this effect by design. For the future, it would be important to understand the role of soft structural disorder in allowing changes of conformations and enhancing the protein promiscuity, and but also, the connection between the observed binding chain reactions with the allosteric regulation effects discussed in disordered proteins in the last years [43, 44]. Finally, our results also suggest that interface prediction methods, such as machine learning predictors, should include some degree of redundancy in the training sets to catch this complexity of the protein interfaces.

## Materials and methods

### Database

We include all protein structures in the PDB [23] (up to the 14/06/2019) available in the PDB file format (which excludes the very large complexes) and with AA sequences of at least 20 residues.

## Interface computation

Protein-protein binding sites are computed using the INTerface Builder method [45] (two AAs are considered in contact as long as their two $C_\alpha$ are at a distance $\leq 5\text{Å}$). INTbuilder is a very fast program to compute protein–protein interfaces, designed to retrieve interfaces from molecular docking software outputs in (an empirically determined) linear complexity. INTBuilder identifies interacting surfaces at both residue and atom resolutions (see https://www.lcqb.upmc.fr/INTBuilder).

To obtain the protein-DNA/RNA binding sites, we looked for the residues whose relative accessible surface area (RASA) decreased upon binding. The change in RASA is computed with naccess [46] (with a probe size of 1.4Å).

## Disorder computation

We have considered two definitions of structural disorder. The first, the missing regions (MR) is formed by the so-called *missing* residues, included in the "REMARK 465" of the PDB structure. The second, the soft disorder that covers all the residues that are poorly determined by the X-ray crystallography study. We identify them by a high B-factor and we consider the B-factor high if the experimental B-factor for its $C_\alpha$ atom is beyond $X$ standard-deviations from the mean B-factor of the entire chain. For this, we used a normalized B-factor, as defined in Eq 1, and imposed the condition $b \geq 1$, with $b$ defined in Eq 1. In general we discuss the case where $X = 1$, but other thresholds (0.5, 2 and 3) are also considered.

## Clustering procedure

We cluster together all the chain structures with similar AA sequences: a minimum of 90% of sequence identity and the same length, up to a 90%. Clusters are created using the MMseqs2 method [47, 48], which is the same method used in the PDB to explore its sequence redundancy, see http://www.rcsb.org/pdb/statistics/clusterStatistics.do for more details. Yet, in contrast to the PDB redundant clusters, we introduced not only a constrain in the protein sequence identity, but also in the sequence length. This means that our condition to cluster chains together is much more restrictive and groups together nearly identical protein chains. The generation script and the full list of clusters are given at www.lcqb.upmc.fr/disorder-interfaces/. The representative chain of the cluster gives the name to the cluster and it is chosen as the one coming from the experimental structure with higher resolution or R-value. Furthermore, we take the sequence and the structure of this representative chain as the reference sequence and structure for the whole cluster. The election of the representative has no noticeable effect in the curves included in this paper (as shown in S13 Fig).

The disordered and binding sites computed on each structure of a cluster are mapped to the cluster's reference sequence via sequence alignment using the Biopython's [49] PAIRWISE2.ALIGN.GLOBALXX routine. A sketch of this process is shown in Fig 3.

The number of different interfaces (NDI) of the cluster refers to the number of such sequences with noticeable distinct interface regions (as for example, the ones we showed in Fig 2). To obtain this number, we extract the minimal group of cluster's structures, so that, all the rest of IRs measured along the cluster are equal in the sequence (up to a 5% of the reference sequence length) to at least one of the interfaces in this minimal group (see a sketch of this extraction in Fig 3). We show the same analysis using a threshold of 1% to define NDI, and if we just plot the results versus the cluster size in S6 Fig.

## Cluster union computation

Once the clusters are constructed, we compute the union over all the MRs, SDRs and IRs observed in the structures of the cluster. More precisely, all the sites that are marked as MR, SDR or IR at least once in the cluster are included in the union of MRs, SDRs and IRs, what we call UMR, USDR and UIR respectively.

## Goodness of the predictor

Later in the manuscript, we will discuss to which extent the USDRs reproduce the UIRs. For this goal, cluster-by-cluster, we consider 5 different estimators of the goodness of this match. These estimators are computed using combinations of the number of true positives (TP, the number of disordered residues that are also IR), true negatives (TN, the number of residues that are neither SDR nor IR), false positives (FP, the number of disordered residues that yet are not IR) and false negatives (FN, those AAs that belong to the interface but were never disordered):

$$\text{Sensitivity (Sen)} = \frac{\text{TP}}{\text{TP} + \text{FN}}, \tag{2}$$

$$\text{Specificity (Spe)} = \frac{\text{TN}}{\text{FP} + \text{TN}}, \tag{3}$$

$$\text{Accuracy (Acc)} = \frac{\text{TP} + \text{TN}}{\text{TP} + \text{FP} + \text{TN} + \text{FN}}, \tag{4}$$

$$\text{Precision (PPV)} = \frac{\text{TP}}{\text{TP} + \text{FP}}. \tag{5}$$

A random prediction of the same exact number of disordered sites, $N_D$, in a chain of $L$ AAs, would predict correctly (in average) an interface site with a probability of $r_I = N_I/L$, being $N_I$ the number of interface sites in the cluster. This means that one expects, also in average ($r_D = N_D/L$), that

$$\text{Sen}^r = r_D, \tag{6}$$

$$\text{Spe}^r = 1 - r_D, \tag{7}$$

$$\text{PPV}^r = r_I, \tag{8}$$

$$\text{Acc}^r = r_D(2r_I - 1) + (1 - r_I). \tag{9}$$

## Disorder predictors

We compared our combined analysis of structural disordered regions with the output of different disorder predictors, in particular the IUPred2A [37, 38] predictor (for the short disorder option), SPOT-Disorder2 [39] and DISOPRED3 [40]. In the case of IUPred2A, and DISOPRED, options to predict disorder-to-order regions are available (ANCHOR and DISOPRED3).

## Supporting information

**S1 Text. Details concerning the supplemental analysis.** We discuss the details of the analysis followed to generate the Supplemental Tables and Figures.
(PDF)

**S1 Fig. Hierarchical organisation of interfaces in the core domain of protein p53 (cluster 4ibs_A).** We reproduce in a larger format Fig 2 in the main text. Cluster 4ibs_A contains 196 structures and 42 different interfaces, among which, 27 are shown here. The rest of them were removed either for lack of space or because they were very similar to the ones shown here. Extra details are given in the S1 Text.
(PNG)

**S2 Fig. Other hierarchical interface organisation. A** for the p73 DNA binding domain (cluster 3vd1_D). Cluster 3vd1_D has 45 structures and 24 different interfaces (the rest of interfaces not shown are hard to distinguish from those displayed). **B** for the C3 exoenzym (cluster 2c8g_A). Cluster 2c8g_A has 47 structures and 10 different interfaces (all shown here). More details are given in the S1 Text.
(TIF)

**S3 Fig. On the normalisation of the B-factor.** We show two columns of figures, the left column is computed using with the normalised b-factor, and the right column using the experimental B-factor. In **A and B**, we show a histogram of the mean value (in each of the set of clusters) of the b-factor of the residues that belong either belong to the DtO or not, and grouped separately if each residue was or not part of the interface for each of the structures of the cluster. Fig **A** shows the same data that Fig 6 in the main-text, but in logarithmic scale. In **C and D**, we show an histogram of the mean value (for each cluster) of the b-factor of the residues that were (or were not) just next to a missing residue in the sequence. In **E and F**, we show the histogram of the b-factor of each of the residues o the cluster representative structure predicted (or not predicted) as disordered by the three predictors considered in the main-text. More details are given in the S1 Text.
(PDF)

**S4 Fig. Cluster representation.** We reproduce the same cluster of Fig 4, but this time showing the partners of each protein chain (in a grey shadow), the binding sites as blue spheres and the b-factor of the chain through a colour code (being deep red very high b-factor and deep blue, very low b-factor). In the centre, we show again the union of all the interface (blue), soft disorder (orange) and disorder-to-order (light blue) regions. More details are given in the S1 Text.
(TIF)

**S5 Fig. Histograms of the cluster composition. A** Bi-dimensional histogram of the number of clusters with a given number of different interfaces and a given size. **B** Number of clusters as function of the number of different interfaces for the groups of clusters that either contain any unbound chain or they do not. **C** Bi-dimensional histogram of the number of unbound chains in the cluster as function of the number of different interfaces. **D** Number of clusters as function of the number of unbound chains contained in each cluster (only clusters with unbound structures considered). More details are given in the S1 Text.
(PNG)

**S6 Fig. Other characterisation of the cluster content.** We repeat some of the curves of the main-text, but this time showing the results of as function of the the number of different interfaces up to the 1% of the sequence (**A** and **C**), or the size of the cluster (**B** and **D**). In **A** and **B**

we show the figures analogous to Fig 8B of the main text, and in **C** and **D**, the figures analogous to Fig 9B. More details are given in the S1 Text.
(PNG)

**S7 Fig. Randomisation tests.** In **A**, we compare the averaged relative size of union of disordered (orange) and interface (blue) regions (shown in Fig 7A in the main-text) with respect to the sequence length as function of the cluster size, with the averaged relative size of the union of fake disordered regions obtained after reshuffling the experimental disordered regions (test 1, red). The randomised disordered regions follow a rather different behaviour with the number of interfaces than the union of experimental interface regions. In **B**, we compare the averaged number of connected disordered regions (DR, orange) and interface regions (IR, blue), normalised by the sequence length, as function of the number of different interfaces in the cluster with the numbers we would obtain if the same number of disordered sites where randomly distributed. We have considered two distinct randomisation tests: a random permutation of the disordered sites in the sequence (test 2) and a reshuffling of the disordered regions but keeping consecutive disordered sites together (test 3). Both tests lead to different curves than the real ones, with the exception of the very big clusters, where the regions superimpose forming a very large cluster. More details are given in the S1 Text.
(TIF)

**S8 Fig. Clusters of high resolution structures.** We compare the PPV (medians by bin) for the USDR($b > 1$) shown in Fig 8B of the main-text (light orange, computed using all the structures of the PDB), with the values we obtain if clusters are just composed of structures with resolution below 2.5Å(dark orange). In dash lines, we show the median expected PPV for a trivial correlation using all structures (black) and structures with high resolution (grey), displaying essentially the same curves. As show, we observe no significant change in the correlation between soft disorder and interfaces with the resolution, despite the fact that curves are now noisier in the high resolution case, because there are less clusters with a high number of structures than in the case studied in the paper.
(PDF)

**S9 Fig. Metrics of the goodness of the match between the USDR and the UIR.** We reproduce the analogous curve to Fig 8A for other possible metrics, including the Sensibility (Sen), the Specificity (Spe), the Accuracy (Acc), the positive prediction value (PPV) and the F1 metrics, which is given by the harmonic mean between the Sensitivity and the PPV. The colour dots correspond to the structures shown in Fig 8C.
(PNG)

**S10 Fig. Metrics excluding the once missing residues.** We repeat Fig 8A but this time excluding from the analysis the residues that are reported as missing at least in one structure of the cluster. The results are indistinguishable from the ones containing DtO residues, thus excluding the possibility that the signal reported is trivially introduced by DtO residues forming interfaces.
(PNG)

**S11 Fig. Progressive change of DRs upon binding. A** We show the change of DRs and IRs in the p73 DNA binding domain (cluster 3vd1_D) along the orange tree branch of S2(A) Fig. In **B**, the change in the C3 exoenzym (cluster 2c8g_A) along the red tree branch of S2(B) Fig.
(TIF)

**S12 Fig. Disorder predictors and soft disorder.** We compare the predictions from the disorder predictors discussed in the main-text, once removed the forever missing residues of the

cluster, with our measures of the USDR. In **A** we show the PPV of the disorder predictions with respect to the different definitions of soft disorder, as function of the NDI. In **B**, we show instead the Sensibility. In **C**, we show the Sensibility versus the Specificity for the different NDI bins.
(PNG)

**S13 Fig. Random choice of the representative chain of the cluster.** We compare the metrics quantifying the agreement of USDR and the UIR shown up to this moment, or if the representative is chosen randomly (instead of by the clustering algorithm). Each cluster is shown as a dot, and the shade of colour indicate the density of points for each bin. We see that the choice of the representative has no systematic effect in the results.
(PNG)

## Acknowledgments

We thank Flavia Corsi and Elodie Laine for useful discussions and technical assistance during early stages of this project.

## Author Contributions

**Conceptualization:** Beatriz Seoane, Alessandra Carbone.

**Data curation:** Beatriz Seoane.

**Formal analysis:** Beatriz Seoane.

**Funding acquisition:** Alessandra Carbone.

**Investigation:** Beatriz Seoane, Alessandra Carbone.

**Methodology:** Beatriz Seoane, Alessandra Carbone.

**Project administration:** Alessandra Carbone.

**Software:** Beatriz Seoane.

**Writing – original draft:** Beatriz Seoane, Alessandra Carbone.

**Writing – review & editing:** Beatriz Seoane, Alessandra Carbone.

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
