## [Decision Letter · Decision Letter 0]

4 May 2020

Dear Dr Seoane,

Thank you very much for submitting your manuscript "The complexity of protein interactions unravelled from structural disorder" for consideration at PLOS Computational Biology.

As with all papers reviewed by the journal, your manuscript was reviewed by members of the editorial board and by several independent reviewers. In light of the reviews (below this email), we would like to invite the resubmission of a significantly-revised version that takes into account the reviewers' comments.

However, we would like to stress that the reviewers have conveyed to us some serious concerns, in particular regarding confusing terminologies and correlations analyses. There are also concerns with the use of the term 'predictor'. The paper would need to be significantly reworked and would need to address all the comments of the reviewers. It would then be subjected to a second round of reviews.

We cannot make any decision about publication until we have seen the revised manuscript and your response to the reviewers' comments. 

Sincerely,

Christine A. Orengo

Associate Editor

PLOS Computational Biology

Arne Elofsson

Deputy Editor

PLOS Computational Biology

Reviewer's Responses to Questions

**Comments to the Authors:**

Reviewer #1: This is a great study with outstanding potential. Although the manuscript is generally well-written, there are sevearl issues that need to be addressed.

1) List of all the abbreviations used in the manuscript should be provided.

2) Reference 1 is wrong. As far as I know, the cited book has only one author, Dr. Peter Tompa.

3) Sentence on lines 20-23 is not clear and should be rephrased: “It is known that IDRs enhance the protein flexibility, and contain a large number of short linear motifs [9], which contributes increase the number of conformational states, the protein’s promiscuity [10–13] and its functional versatility [14, 15].”

Sentence on lines 28-29 should be reworded for clarity “In addition, IDRs often structure after the binding, suffering a so-called disorder-to-order transition [10], …” I think that it should be something in line with: “In addition, IDRs often become at least partially structured after the binding, undergoing a so-called binding-induced disorder-to-order transition [10], …”

4) Sentence in lines 68-69 is not clear “This extended definition of disorder gathers, at the same time, the flexible and the amorphous parts of the chain”. What are the amorphous parts of the chain? Should they be “ambiguous” (i.e., showing different structure in different PDB entries of similar proteins)?

5) The authors considered a protein chain as bound if an interface can be measured with at least one other protein/DNA/RNA chain in the PDB complex, whereas the protein chain was classified as unbound otherwise. How about the presence in a PDB structure of small molecules and not the other protein/DNA/RNA chains? In my view, protein-based complexes should include both types of interactions – binding of other protein/DNA/RNA chains and binding of small molecules. It is well-known that small molecule binding can dramatically affect protein structure. Therefore, it would be very important to conduct such an analysis and compare the results of the analysis of these two different types of protein complexes. However, I recognize that this represents an entirely new project that probably should be a subject of subsequent study. On the other hand, I also think that a brief discussion of this issue should be added to the manuscript.

6) Some aspects related to the importance of intrinsic disorder to assembly of proteinaceous machines were discussed in PMID: 24702702. Among various disorder-based mechanisms related to the assembly of protein complexes, a stepwise targeting and directional sequential assembly mechanism (binding chain reaction) was proposed there (figure 3 in that paper), where binding-induced (partial) folding of an IDP/IDR can generate a new conformation with a novel binding site for a new partner. In my view, this stepwise assembly mechanism resembles the mechanism of protein complex assembly described in the current manuscript.

Probably, the authors should also comment on the correlation between intrinsic disorder-based assembly and allosteric regulation.

There are several linguistic and stylistic issues in the current version of the manuscript. Therefore, the manuscript should be carefully edited by native English speaker or by a professional editor.

Reviewer #2: The manuscript by Seoane & Carbone describes the correlation between intrinsic disorder and interaction interfaces as calculated on a clustered version of the Protein Data Bank (PDB) from June 2019. In its present form, the paper appears problematic for various reasons:

1. Terminology. The authors seem to use non-standard, somewhat cereless or confusing terminology, e.g. referring to "intrinsic disorder" (which they largely use themselves) as "structural disorder" in the title and some other places. Likewise, when clustering their PDB proteins they refer to "homologous families of proteins" to groups with > 90% identity and > 90% coverage (not "length") which are essentially the same protein. Unfortunately, a number of these issues detract considerably from the clarity and understandability of the manuscript.

2. The authors pack together missing PDB residues and residues with B-factor > 1 standard deviation above average for the chain as "disordered". This reviewer has to take issue as there is no statistical analysis on the relative abundance of the latter vs. the former. In particular, this construction is tautological when applied to "predict" interactions (see next point). Moreover, it cannot be said that residues with a B-factor of, say, 30 when the average is 20 and SD 5 can be considered "disordered" in a structure. Quite far from it.

3. Circular argument for missing residues. The authors look at large clusters, where many PDB structures correspond to the same (or very similar) protein solved with different binding partners. They use this set to establish residues which are missing in structure A, but solved in structure B. Knowing how crystallography works, it is trivial to say that the same protein alone would be crystalized in the same way (i.e. same missing residues) over and over again. In many cases the crystallographer would even eliminate possibly disturbing parts from the construct to help crystallization. However, the author's method only picks up these residues as "disordered" mainly because there is a different strcuture where the coordinates are present. Hence, the new structure "proves" both "disorder" and "interaction"... neither of which exists in absence of the other.

4. Flawed argument for "high B-factor" residues. The point above becomes worse for high B-factor residues. In this case, structure B will have lower B-factors compared to A precisely because there is a binding partner increasing rigidity. At this point the correlation becomes meaningless.

5. Key observation as "predictor". Even assuming that there is a value to the observation, which this reviewer does not believe, it is not possible to sell this as a "predictor". Without multiple structures, the method will not be able to "predict". With multiple structures, the "prediction" is trivial and the performance metrics meaningless. Again, this may be due to a careless terminology. The authors should write about analysis rather than prediction.

6. Several statements, especially in the abstract and discussion, have to be toned down in light of the points above.

Specific points:

* The text should be re-written for simplicity and clarity. Many sentences are quite winding, with qualifying statements detracting from the logic of the argument.

* Clustering. The authors should try to use SIFTS and UniProt accessions, available from PDBe, to use a standard procedure.

* Clarify PDB "old format".

* The "INTerface builder" method should be better explained.

* Figures will benefit from clearer captions. E.g. in Figure 1, why is "2bio_A" in the center, yet the legend talks about "4ibs_A"?

* The order of figures does not match the text.

* "Results" appears twice s a section heading.

Reviewer #3: The study, while certainly interesting in design, sticks rigidly to its scope, when an obvious extension (particularly given the conclusions) would be to compare results using a few disorder predictors. The findings in the study do not tend to support the conclusions, and negative results displayed in the figures are glossed-over in the text. Overall, I believe that there are a number of issues that need to be addressed before it would be ready for publication. It would also greatly benefit from a round of grammatical revision.

Issues:

• There are reasonable citations to support the work being done, but insufficient citation of previous similar works (e.g. PMC2646137) nor a comparison of conclusions.

• At a number of points, it is mentioned that computational disorder predictors don’t take into account all information, thus are relied upon to generalize. An actual comparison of the predicted disorder (from a couple of better known predictors) to disorder defined in this paper would be useful. A comparison of results between using predicted and measured disorder to define interfaces would be even better!

• How is the reference sequence for each cluster determined? Do the results change much if a different (randomly selected?) member of the cluster is used as the reference sequence?

• As well as ‘missing’ residues, any residues with a B-factor greater than 1SD from the chain mean is classed as disordered. However, this means that around 1/3 of every protein is defined as disordered regardless of the quality of the crystal structure. (2/3 of the data should lie within 1SD of the mean). What would be the result if a static threshold were used instead?

• “We consider two interfaces as different if the number of non-mutual AAs in the interfaces is larger than 5% of the sequence length. This means that two interfaces that are concentric or slightly displaced, but one significantly larger than the other, are counted as different interfaces.” Does this approach introduce a size bias due to surface-to-volume ratio? Is 5% not too much? I consider a 100 AA protein where 15 residues are interface. This says 5 residues can be different (1/3 of the interface) and it is still considered the same interface. What happens when this threshold is lowered?

• “lead different curves, as shown in Fig. S3.” This actually refers to Figure S2B. Also, while the full randomisation gave a distinctly different curve (more connected regions at lower NDI) the site randomisation was actually very similar to observed DR curve.

• So, to clarify: your predictor is literally: if it is disordered in the UDR: it is predicted to be an interface in the UIR? The text describes the DR being used to predict the IR, which isn’t possible since it was stated earlier that they rarely occur in the same place. I think you mean UDR and UIR.

• Is an estimator the right term for a prediction quality score? It is at best, misleading: implying that these are estimates of interface propensity rather than quality of interface predictions.

• The Bio.pairwise2.align.globalxx() function has no gap penalties. While this is much less of an issue with limits on sequence identity and length, it can still insert gaps into the reference sequence. How is it handled when a reference sequence gap is labelled as DR/IR?

• Figure 1. An interesting and well-constructed figure, however the text overlaying the radial lines hinders reading it (the radial lines can probably be removed, and simply state that some complexes were excluded).

• The x-axis of Figure 3A should be ‘number’ rather than ‘size’

• With all the logarithmic bar charts / box plots it would be nice to have some graphical indication of how bin size changes on the logarithmic axis.

• Figure 5. Performance of this predictor looks considerably worse than is described in the text. It is essentially worse-than random guessing for predicting interfaces when the NDI is <10 (which covers the majority of clusters). Instead of presenting these 5 different metrics, a ROC curve, or a PR curve would give a better idea of method performance.

• Figure 8A. While the disordered regions do bind, these are also some disordered regions which never bind (the lower loops of the 5clx complex). Is there an explanation for these?

• Figure 6 C and D are never refered to in the text. I believe that references in the text to figure 6B actually refer to figure 6D, while B and C should be part of A.

• The discussion is very vague and simply re-iterates the abstract. It touches once more on computational disorder prediction methods and how they can be improved greatly, however never compares results from them. Given that pathological implications were touched on in the introduction, this should also be expanded upon here. Just further discussion about the implication of the results in general.

**Have all data underlying the figures and results presented in the manuscript been provided?**

Reviewer #1: Yes

Reviewer #2: Yes

Reviewer #3: None

PLOS authors have the option to publish the peer review history of their article (what does this mean?). If published, this will include your full peer review and any attached files.

Reviewer #1: Yes: Vladimir N. Uversky

Reviewer #2: No

Reviewer #3: No
---

## [Decision Letter · Decision Letter 1]

22 Sep 2020

Dear Dr Seoane,

Thank you very much for submitting your manuscript "The complexity of protein interactions unravelled from structural disorder" for consideration at PLOS Computational Biology.

As with all papers reviewed by the journal, your manuscript was reviewed by members of the editorial board and by several independent reviewers. In light of the reviews (below this email), we would like to invite the resubmission of a significantly-revised version that takes into account the reviewers' comments.

In particular one of the reviewers has highlighted serious flaws that would need to be addressed particularly around the identification of disordered residues and the use of nomenclature like 'soft disorder' not widely used by the IDP community.

We cannot make any decision about publication until we have seen the revised manuscript and your response to the reviewers' comments. Your revised manuscript is also likely to be sent to reviewers for further evaluation.

Sincerely,

Christine A. Orengo

Associate Editor

PLOS Computational Biology

Arne Elofsson

Deputy Editor

PLOS Computational Biology

Reviewer's Responses to Questions

**Comments to the Authors:**

Reviewer #1: In my view, all the critiques were adequately addressed and the manuscript was revised accordingly.

Reviewer #2: The manuscript by Seoane & Carbone has been extensively modified. Unfortunately, this reviewer has not been swayed by the authors' "exercise to prove them wrong".

1. X-ray crystallography. The authors seem oblivious to the problems arising from different resolutions and crystal quality. What they attribute to "disorder", i.e. the difference in missing residues and/or high B-factors, does not consider diffraction quality as a much more obvious answer.

2. Non-standard nomenclature and jargon. The authors have not eliminated the use of many non-standard terms. This denotes a lack of attunement to the scientific community they are trying to address. In particular, this reviewer takes issue with the term "soft disorder" in the way it has been conceived. Again, any crystallographer would probably call it just "flexible residues".

3. Circular argument. This reviewer stands by its previous remarks (see points 2-4 from the previous review) and has not been convinced.

4. High B-factors. As also pointed out by reviewer #3, this definition for "soft disorder" is flawed and inflates the perceived fraction of "disorder" for perfectly rigid structures.

5. Flexibility. In general, the authors seem very convinced that they have found an important concept and try to promote it in several ways which do not appear consistent with the existing literature. While "disorder" is a trendy topic, not everything is best explained with this concept. The decades old notion of flexibility is much better suited to several key observations of this manuscript.

Specific points:

* Fig. 1. This reviewer did not find it very useful to explain the concepts as it does not address the definitions themselves in a graphical way.

* The reference for DisProt is outdated.

* Clustering. The authors have brushed away the request to use SIFTS and UniProt accessions because their method "works perfectly". The request weas not for a "better" procedure but for a "reproducible" one.

Reviewer #3: I am happy that the authors have adequately addressed my comments.

**Have all data underlying the figures and results presented in the manuscript been provided?**

Reviewer #1: Yes

Reviewer #2: Yes

Reviewer #3: None

PLOS authors have the option to publish the peer review history of their article (what does this mean?). If published, this will include your full peer review and any attached files.

Reviewer #1: **Yes: **Vladimir N. Uversky

Reviewer #2: No

Reviewer #3: No
---

## [Editor Report · Decision Letter 2]

18 Nov 2020

Dear Dr Seoane,

We are pleased to inform you that your manuscript 'The complexity of protein interactions unravelled from structural disorder' has been provisionally accepted for publication in PLOS Computational Biology.

Best regards,

Christine A. Orengo

Associate Editor

PLOS Computational Biology

Arne Elofsson

Deputy Editor

PLOS Computational Biology

---

## [Editor Report · Acceptance letter]

30 Dec 2020

PCOMPBIOL-D-20-00227R2 

The complexity of protein interactions unravelled from structural disorder

Dear Dr Seoane,

I am pleased to inform you that your manuscript has been formally accepted for publication in PLOS Computational Biology. Your manuscript is now with our production department and you will be notified of the publication date in due course.

With kind regards,

Livia Horvath
